# Range Dynamics of Striped Field Mouse (*Apodemus agrarius*) in Northern Eurasia under Global Climate Change Based on Ensemble Species Distribution Models

**DOI:** 10.3390/biology12071034

**Published:** 2023-07-22

**Authors:** Varos Petrosyan, Vladimir Dinets, Fedor Osipov, Natalia Dergunova, Lyudmila Khlyap

**Affiliations:** 1A.N. Severtsov Institute of Ecology and Evolution of the Russian Academy of Sciences, Moscow 119071, Russia; osipov_feodor@mail.ru (F.O.); nndergunova@gmail.com (N.D.); khlyap@mail.ru (L.K.); 2Psychology Department, University of Tennessee, Knoxville, TN 37996, USA; dinets@gmail.com

**Keywords:** *Apodemus agrarius*, rodent, climate change, global circulation models, invasive species, species distribution models, eSDM, zoonotic pathogens

## Abstract

**Simple Summary:**

Global climate change may expand the regions suitable for alien species beyond their historical range. The striped field mouse (*Apodemus agrarius*) is a widespread species in Northern Eurasia. Its current and future range expansion under climate change may negatively influence public health and the economy. We studied the potential distribution of the striped field mouse and assessed vulnerability of Northern Eurasia to *A. agrarius* invasion. We created an ensemble of species distribution models to predict suitable niches across current and future climate changes. We found that the range changes depended on both the sensitivity and scenario of climate change models. The main trends included range expansion to the northeast, partial habitat loss in the steppe, and formation of a continuous range from Central Europe to East Asia. The results are important for minimizing new invasions of the striped field mouse and their negative consequences.

**Abstract:**

The striped field mouse (*Apodemus agrarius* Pallas, 1771) is a widespread species in Northern Eurasia. It damages crops and carries zoonotic pathogens. Its current and future range expansion under climate change may negatively affect public health and the economy, warranting further research to understand the ecological and invasive characteristics of the species. In our study, we used seven algorithms (GLM, GAM, GBS, FDA, RF, ANN, and MaxEnt) to develop robust ensemble species distribution models (eSDMs) under current (1970–2000) and future climate conditions derived from global circulation models (GCMs) for 2021–2040, 2041–2060, 2061–2080, and 2081–2100. Simulation of climate change included high-, medium-, and low-sensitivity GCMs under four scenarios (SSP1-2.6, SSP2-4.5, SSP3-7.0, and SSP5-8.5). We analyzed the habitat suitability across GCMs and scenarios by constructing geographical ranges and calculating their centroids. The results showed that the range changes depended on both the sensitivity of GCMs and scenario. The main trends were range expansion to the northeast and partial loss of habitat in the steppe area. The striped field mouse may form a continuous range from Central Europe to East Asia, closing the range gap that has existed for 12 thousand years. We present 49 eSDMs for the current and future distribution of *A. agrarius* (for 2000–2100) with quantitative metrics (gain, loss, change) of the range dynamics under global climate change. The most important predictor variables determining eSDMs are mean annual temperature, mean diurnal range of temperatures, the highest temperature of the warmest month, annual precipitation, and precipitation in the coldest month. These findings could help limit the population of the striped field mouse and predict distribution of the species under global climate change.

## 1. Introduction

In recent years, research on climate change and biological invasions has intensified worldwide [1,2,3,4]. Studies have shown that climate change affects disease transmission and other aspects of the infectious process [5,6,7,8], for example, blood-sucking vectors [9,10]. A major role of climatic factors in the circulation of pathogens of hantavirus infections [11] and leptospirosis [12] involving invasive small mammals has been identified [13]. In the case of the brown (*Rattus norvegicus*) and black (*R. rattus*) rats [14], it has been shown that rats widely dispersed pathogens (particularly plague) from their native ranges in the process of their invasions. In addition, as rats have invaded new territories, the number of pathogens found in them has increased at least threefold [14]. A global meta-analysis of changes in the range boundaries for 1700 species in the Northern Hemisphere showed that the northern boundaries moved 6.1 km per decade northward, or 6.1 m per decade upward (*p* < 0.02) [15]. These estimates were developed for a moderate temperature increase of 0.74 °C in the 20th century, which caused latitudinal and altitudinal range shifts for many species [16]. Consequently, with the accelerated temperature increase predicted for the 21st century, we expect even greater shifts in the species ranges. This will affect the extensiveness of the invasive process and the possible increase in ecological and economic damage from invasive species [17,18].

Climate change, including extreme events (e.g., floods, fires), can accelerate invasive processes [19,20,21], and have a profound impact on the environment. In addition, human impacts of transcontinental travel, land degradation, and conversion of natural ecosystems to farmlands lead to the spread of many alien species, including the striped field mouse (SFM)—*Apodemus agrarius* (Pallas, 1771) [21,22]. In Russia, the SFM carries 15 human pathogens [23,24], and the list of diseases continues to grow [25,26]. The SFM is a good host of ectoparasites, but it plays the greatest role in such non-vector-borne diseases as hemorrhagic fever with renal syndrome (HFRS), leptospirosis, and tularemia [27]. The *Hantaan orthohantavirus*, whose main host is the SFM inhabiting the Far East of Russia, causes the most severe course of the disease in humans, with a high mortality rate [28,29]. In the European part of the range, another hantavirus, less dangerous for humans, is associated with the SFM: *Dobrava-Belgrade orthohantavirus*, genotypes *Kurkino* and *Saaremaa* [30,31,32]. Among the leptospira, the SFM is the main host of *Leptospira kirschneri*, serovar *Mozdok*, whose active foci are located in North Ossetia [23,33], and are found in Germany [34]. Extensive invasions of the past, ongoing range expansion, and tangible negative consequences for public health and agriculture [35] justify including the SFM in the list of the most dangerous invasive species of Russia [36].

The SFM has undergone a significant range expansion over the past 24 kyr., with the most recent invasions occurring in the 20th century. Currently, the striped field mouse is a Palearctic rodent with a range divided into two large isolated areas: the western part (Europe–Siberia–Kazakhstan) and the eastern part (Far East-China) [36,37,38]. The significant part of its modern range is located in Russia. According to molecular genetic analysis, the eastern part of the range is more ancient, while invasion to western regions of Eurasia occurred later [38,39,40,41]. In the postglacial prehistoric times (about 17.7 (95% HPD 13.2–22.5) kyr.) [41], the SFM occupied West Siberian and East European habitats with floodplain forests, shrubs, and meadows [23,42,43,44,45,46]. In the preagricultural times, the mouse distribution was mosaic [47]. The SFM adapted to disperse widely, change habitats, rapidly build up their numbers, and huddle in the secondary habitats during winter [48]. These features are characteristic of many invasive species [49], and they fully manifested with the beginning of anthropogenic transformations of landscapes, starting with plowing. Striped field mice emerged from secondary habitats and invaded the croplands, field margins, and arable lands [44,45,46,50]. By the 19th century, alongside the plowing of the forest-steppe area, the SFM had taken over its current range in Eastern Europe [36,49,51], then moved north (taiga area) with deforestation for agriculture, and followed irrigation projects down south (steppe area) [23,44,51]. The northward expansion was mainly associated with human settlements [52,53]. During the second half of the 20th century, due to global climate changes, the SFM range expansion was noted in Germany, Italy, Austria, Czech Republic, Slovakia, Hungary, Slovenia, Ukraine, Moldova, Azerbaijan, and Kyrgyzstan [14,15,33,34]. The invasion of the species continued at the turn of the 20th and 21st centuries [36,37,41].

Spatial distribution models (SDMs) are powerful modelling tools for analyzing invasion risk and predicting potential range under climate change using (1) mechanistic SDMs based on functional traits and physiological constraints, and (2) correlative SDMs relating occurrence data to spatial environmental data [54,55]. There are case studies showing that the two methods give consistent predictions both under current and different climate change scenarios [55]. However, correlative SDMs are relatively more used for predicting the spatial distribution of species and the effects of climate change because they have important advantages over mechanistic modelling methods. The advantages of the correlative methods include the simplicity and flexibility of their data requirements, their relative ease of use within freeware packages, and the ability to account for interactions between different environmental factors. In this study, we use correlative SDMs to build models of the species under current and feature climate change conditions. There are several examples of the successful application of ecological modelling tools for biogeographic analysis aiming to construct species distribution models for the TOP-100 most dangerous invasive species in Russia [56]. However, these studies only included invasive trend analysis and SDM construction under current climate conditions.

The goal of this study was to assess the impact of climate change on the distribution of the SFM in northern Eurasia, and predict areas which may be suitable for establishment under current and future climate scenarios. First, we used ensemble predictions extracted from seven individual species distribution models (iSDMs), using 12 global climate change models from the Coupled Model Intercomparison Project 6, grouped into three types (high-, medium-, and low-sensitivity models), and four scenarios to predict suitable climatic regions for the SFM. Second, using these ensemble models, we created maps of suitable regions for current (1970–2000) and future (2021–2040, 2041–2060, 2061–2080, and 2081–2100) periods. Third, using predicted range changes, we assessed the vulnerability of Northern Eurasia to SFM invasion.

## 2. Materials and Methods

### 2.1. Species Occurrence Records

We created the database of species occurrence records (hereafter records) in ArcGis Desktop 10.8.1 [57] using our original field data [38], together with data from international open access sources (GBIF) (www.gbif.org, accessed on 3 April 2018) (https://doi.org/10.15468/fv3hn3, accessed on 24 May 2021, https://doi.org/10.15468/dl.zmubis, accessed on 3 April 2018). We distinguished three types of records. The first type contained records described by exact geographic coordinates. For this data type, we removed duplicates and applied accuracy control filters (i.e., records with location accuracy greater than 5 km were excluded). The second type included records with locations depicted in the maps without accurate coordinates. For this type of record, we determined geographic coordinates from the base maps through geo-registration and linking the available locations to base maps using at least 30 control points selected in the ArcGis Desktop 10.8.1 environment. Base maps were obtained from Natural Earth public domain map datasets (https://www.naturalearthdata.com/downloads/10m-cultural-vectors, accessed on 3 June 2018 ). For the third type of data, we used only records, which allowed us to determine the exact geographic coordinates using GoogleEarth (https://earth.google.com, accessed on 11 July 2018 ) with an accuracy of at least 5 km. We obtained the final dataset of 2012 records by eliminating duplicate records and combining all three types of records.

### 2.2. Environmental Data (Current Climate)

In our study, we used 19 bioclimatic variables (hereafter variables) averaged over the period 1970–2000, with a spatial resolution of 2.5 arc minutes (~5 km × 5 km), from the WorldClim 2.1 database (https://www.worldclim.org/data/worldclim21.html, accessed on 11 July 2018) [58]. Current (1970–2000) variables were estimated based on monthly minimum, average, and maximum values of temperature and precipitation. The variables used in our study included Bio_01 (annual mean temperature, °C), Bio_02 (mean diurnal range (mean of monthly (max temp–min temp) °C)), Bio_03 (isothermality (Bio_02/Bio_07) × 100, %), Bio_04 (temperature seasonality (standard deviation × 100), °C), Bio_05 (max temperature of warmest month, °C), Bio_06 (min temperature of coldest month, °C), Bio_07 (temperature annual range (Bio_05–Bio_06) °C), Bio_08 (mean temperature of wettest quarter, °C), Bio_09 (mean temperature of driest quarter, °C), Bio_10 (mean temperature of warmest quarter, °C), Bio_11 (mean temperature of the coldest quarter, °C), Bio_12 (annual precipitation, mm), Bio_13 (precipitation of the wettest month, mm), Bio_14 (precipitation of the driest month, mm), Bio_15 (precipitation seasonality (coefficient of variation), mm), Bio_16 (precipitation of the wettest quarter, mm), Bio_17 (precipitation of the driest quarter, mm), Bio_18 (precipitation of the warmest quarter, mm), and Bio_19 (precipitation of the coldest quarter, mm). Previous studies showed the high efficiency of this set of variables for building species distribution models under current and future global climate changes [56,59,60,61,62,63,64].

### 2.3. Thinning Records and of Environmental Variables

To identify and reduce the spatial autocorrelation of records, we used a two-step process [56]. First, we generated 17 subsamples of records in the *spThin* R package [64] by using 17 thinning parameters with minimum distances between points from 35 to 595 km in intervals of 35 km. Then, these 17 datasets were subjected to cluster analysis through the average nearest neighbor index (ANNI) in ArcGis 10.8.1. The ANNI metric estimates the degree of clustering of the records, with an estimate of the ratio of the average distance from each point to its nearest neighbor to the expected average distance for a random distribution. If ANNI = 1, the distribution is random; if ANNI > 1, the distribution is dispersed; if ANNI < 1, the distribution is clustered. After this analysis, we selected a reduced set of geo-referenced records DS_red_ for which ANNI = 1. 

A two-stage procedure was also used to select variables from the full WorldClim 2.1 dataset. First, we determined the training area of the iSDM by using the conventional choice of selecting minimal convex polygons, which contained DS_red_ [65]. However, the conventional choice of background localities in convex polygons may have some limitations, as this minimizes the contrast between presence and absence, so we also employed the recommendations provided in the previous literature [66,67]. These studies suggest selecting backgrounds from areas that are immediately adjacent to occupied habitats, but are known to be unoccupied. For this reason, we combined two convex polygons located in the western (Europe–Siberia–Kazakhstan) and eastern (Far East-China) parts of the species’ range, which directly included both occupied and unoccupied areas. Second, all records from the DS_red_ and the background points from the training area were represented in the multidimensional space of variables (Bio_01, Bio_02, ..., Bio_19). Next, we used principal component analysis (PCA) with the “ade4” R package to visualize the points in the plane formed by two PCA axes, i.e., we obtained a graphical representation for: (a) distribution of background points, and plotting the scattering ellipsoid of records; and (b) representation of the correlation between climatic variables with the construction of a correlation circle [55]. The obtained graphical objects (ellipsoid, correlation circle) allowed us to select the first two variables. The variable from the correlation circle that is parallel to the major axis of the scattering ellipsoid is chosen as the first variable. The second variable is the one that is parallel to the minor axis. The remaining variables from the dataset(Bio_01, Bio_02, …, Bio_19) were selected for inclusion in the models using the *ENMTools* R package [68]. Those variables were excluded, between which, the Spearman’s pairwise rank correlation coefficient was greater than 0.72 in absolute value. Multicollinearity was assessed by the VIF (variation inflation factor) using the *Usdm* R package [69]. A predictor variable is considered multicollinear and is excluded if VIF > 5 [55].

### 2.4. Environmental Data (Future Climate)

Distributions of the SFM were also modeled for future climatic conditions. To assess the potential impact of global climate change on range dynamics of *A. agrarius*, we analyzed 40 global circulation models (GCMs) from the Coupled Model Intercomparison Project (phase 6) (CMIP6) for the various climate change scenarios (Shared Socioeconomic Pathways, SSPx-y): SSP1-2.6 (low greenhouse gas (GHG) emissions in which CO_2_ emissions are reduced to zero around 2075), SSP2-4.5 (intermediate GHG emissions in which CO_2_ emissions increase around the current rate until 2050, and then decrease but not reach net zero by 2100), SSP3-7.0 (high GHG emissions where CO_2_ emissions double by 2100), and SSP5-8.5 (very high GHG emissions where CO_2_ emissions triple by 2075) [70,71]. The CMIP6 GCMs account for physical, chemical, and biological processes more accurately than the models used in the CMIP5 [72,73,74]. Forty CMIP6 GCMs predict equilibrium climate sensitivity (ECS) values ranging from 1.83 to 5.67 °C [75,76]. 

The ECS is the most important climatic parameter. It is defined as the change in global average surface air temperature resulting from a doubling of carbon dioxide (CO_2_) concentration as soon as the associated ocean–atmosphere–sea ice system reaches equilibrium. To account for the main trends of climate change and variability of predictions, we divided all CMIP6 GCMs into three groups characterized by different ESC values and selected 12 GCMs, as recommended in the literature [75,76]. These 12 GCMs are divided into three groups: high-sensitivity models (Hsens) GCMs—CanESM5 [77], CNRM-CM6 [78], CNRM-ESM2-1 [79], and IPSL-CM6A-LR [80] (4.6 ≤ ECS ≤ 5.6); medium-sensitivity models (Msens) GCMs—CNRM-CM6-1-HR [81], EC-Earth3-Veg [82], MRI ESM2-0 [83], and BCC-CSM2-MR [84] (3.0 ≤ ESM ≤ 4.3), and low-sensitivity models (Lsens) GCMs—MIROC-ES2L [85], MIROC6 [86], GISS-E2.1 [87], and INM-CM4.8 [88] (1.8 ≤ ESM ≤ 2.7). For the selected models (12) and scenarios (SSP1-2.6, SSP2-4.5, SSP3-7.0, and SSP5-8.5), raster layers were created in *.gtif format, including variables (Bio_01, ..., Bio_19) for the time periods (2021–2040, 2041–2060, 2061–2080, and 2081–2100) under four SSPx-y scenarios. We created three sets of ensemble models (Hsens GCMs, Msens GCMs, and Lsens GCMs) by taking average values, and used the ensemble values as predictors. Our multimodel ensembles average accounts for inherent variability of different GCMs and future climate change scenarios.

### 2.5. Species Distribution Modelling

In recent years, different methods, including regression, machine-learning, and classification, have been used to create SDMs [56,61,89,90]. Discrepancies between methods can be significant, and the performance of algorithms can differ [56,91]. Considering the variability between methods and algorithms, we decided to use the ensemble modelling approach. Ensemble models that combine predictions from multiple habitat models (called ensemble members or individual models—iSDMs) may demonstrate higher performance and robustness [56,91,92,93,94]. By using a wide range of approaches, ensemble modelling can account for intermodel variability and uncertainty in projections [93]. However, prediction uncertainty may also depend on GCMs and SSPx-y variation, which is important for understanding basic trends [93]. 

We used the seven most effective iSDM building methods [56,90,93], which are implemented in the *Biomod2* v.4.2 R package [95]. They include two regression methods (GLM: generalized linear model, GAM: generalized additive model), four machine-learning methods (ANN: artificial neural network, GBM: generalized boosting model, RF: random forest, and MaxEnt: maximum entropy method), and one classification method (FDA: flexible discriminant analysis) [95].

#### 2.5.1. Determination of the iSDM Optimal Parameters 

Although the tuning parameters of seven iSDMs (GLM, GAM, GBM, FDA, RF, ANN, and MaxEnt), defined by default in the *Biomod2* package [90], are based on a large set of empirical data, they are not always effective [56,95,96]. For this reason, we have determined the optimal parameters for each of the iSDMs using the *Biomod2_tuning* function. Determining these parameters usually requires (in single-threaded-sequential-computing mode) a large amount of computer time. To avoid this, we took advantage of the multithreaded computation mode (i.e., parallel computation) in the R package *doParallel* [97]. The *Biomod2_tuning* function provides two metrics to determine optimal iSDM parameters: AUC and AIC. The area under the curve (AUC) allows for the evaluation of the quality of the binary classification and the effectiveness of prediction, i.e., the AUC value characterizes the predictive power of a model [98]. According to the AUC value, the model was graded as “poor” (if AUC = 0.6–0.7), “satisfactory” (AUC = 0.7–0.8), “good” (AUC = 0.8–0.9), or “excellent” (AUC = 0.9–1.0) [98]. The Akaike metric (AIC) is a widely used measure of complexity (number of explanatory variables used) and log-likelihood of a model. Among the models built, the one for which the AIC value is minimal is the best for specific data [98]. For some models (including GLM, GAM, and MaxEnt), the optimal parameters are determined by minimizing AIC. For others, the optimal parameters are determined by maximizing the AUC. A description of the basic methods and algorithms that were used to determine the optimal parameters of iSDMs is presented in the Appendix A.

#### 2.5.2. Building iSDMs and Evaluating Model Performance

We constructed iSDMs using seven methods (GLM, GAM, GBM, RF, FDA, ANN, MaxEnt) implemented in the *Biomod2* v.4.2 R package [89]. For all iSDMs, we generated 1000 pseudoabsence points in the training area. The number of pseudoabsence points was generated [99] according to the number (N) of records (if N ≤ 1000 then 1000 points were selected, else 10,000 were selected). This procedure was repeated three times. The pseudoabsence points were obtained using the “Sre” generation strategy [95], because this strategy reduces the random generation of false pseudoabsence points. Individual models (iSDMs) were then constructed using the *BIOMOD_Modeling* function with the optimal parameter values identified in the previous step (see Section 2.5.1). Optimal parameters in *Biomod2* were set using the function *BIOMOD_ModelingOptions*, i.e., all default parameters for all models were replaced by their optimal values. The predictive performance of each iSDM was evaluated by randomly splitting the records three times into two parts, i.e., the models were trained using 80% records, and the accuracy was evaluated using the remaining 20% of records. For each of the seven methods, nine iSDMs were created for three different sets of pseudoabsence points, and three different runs of dividing the records into two parts of samples for training and testing. The predictive performance of the iSDMs was assessed using three metrics: true skills statistics (TSS), Cohen’s Kappa (KAPPA), and AUC [98]. The TSS metric ranges from −1 and +1, where +1 indicates “perfect” agreement, “excellent” 0.8 < TSS < 1, “good” 0.6 < TSS ≤ 0.8, “satisfactory” 0.4 < TSS ≤ 0.6, and “poor” TSS ≤ 0.4 [55]. The KAPPA metric assesses model predictive performance on the following scale: “excellent” KAPPA > 0.75, “good” 0.4 < KAPPA ≤ 0.75, and “poor” KAPPA ≤ 0.40 [55].

For each iSDM, we also created response curves and the relative importance of each variable [89]. In general, it is difficult to assess the importance of predictor variables because they are not based on the same algorithms, methods, and approaches. However, the *Biomod2* package provides a common approach for estimating a measure of the importance of each variable, which is independent of the model. Once iSDMs with optimal parameters are constructed (i.e., there is a standard or reference predictions), one particular variable is randomized and a new prediction model is created. Then, the Pearson correlation coefficient (Pcor) between this new prediction and the reference prediction is calculated, and this Pcor coefficient is considered to give an estimate of the importance of the variable in the model. The importance of a variable is determined by the formula: VarI = 1 − Pcor. This means that if there is a high correlation between the two predictions, then the randomized variable has little effect on the prediction, and it is considered not important to the model. In contrast, a low correlation (Pcor close to 0) indicates a significant difference in prediction, indicating the importance of this variable to the model. It should be noted that this method takes into account only the direct effects of the variables, and does not allow us to identify the combined effect of the importance of the variables. 

#### 2.5.3. Building Ensemble Models (eSDMs) under Current and Future Climate Conditions

**Ensemble modelling under current climate conditions.** To ensure that the ensemble models did not use spurious models with low metrics of predictive performance, we included only those iSDMs for which the TSS metric was above 0.8 (see Section 2.5.2). In our study, two relevant methods of ensemble aggregation were used—committee averaging (CA), and weighted mean (WM) [55,89]. In addition, we determined the coefficient of variation (CV) of the models in the ensemble, which informs us about the extent of agreement or divergence of predictions between models. These estimates allow us to identify the areas where the predictions of the iSDMs diverge most. Since it is not known a priori which method of combining iSDMs for building an eSDM is better, we first built three types of models (eSDM_CA_, eSDM_WM_, and eSDM_CV_). Then, to obtain the final eSDMs, we evaluated the model predictive performance of each method of aggregation using TSS metrics, and selected the best one with the highest TSS value for projection from the training area into Northern Eurasia. 

For additional checking of the predictive performance of eSDMs, we also used the Boyce index (B_ind_) [100,101]. The B_ind_ only requires records, whereas KAPPA, AUC, and TSS require both records and pseudoabsence points. The Boyce index measures how much the predictive models differ from random distribution only using records. This index varies from −1 to 1. Positive values indicate that the predictive model is consistent with the occurrence data, values close to zero mean that the model does not differ from random distribution, and negative values provide evidence of counter predictions, i.e., predict poor-quality areas where presences are more frequent [101]. We calculated the B_ind_ for each of the 10 model replicates, and the averaged its estimates. The reasonableness of using this index has been shown in previous studies [56,62,101,102,103]. 

**Ensemble modelling under various group of GCMs and scenarios.** Within the concept of ensemble modelling in Biomod2, we used constructed iSDMs with optimal parameters and determined the best method of their combination to build eSDMs for different groups of GCMs and scenarios of climate change in time. As a result, we built 48 ensemble models for three groups of CMIP6 GCMs (Hsens, Msens, Lsens) and four climate change scenarios (SSP1-2.6, SSP2-4.5, SSP3-7.0, and SSP5-8.5) with 20-year steps (2021–2040, 2041–2060, 2061–2080, and 2081–2100).

### 2.6. Assessment of Range Dynamics under Future Climate (Various Groups of GCMs and Scenarios)

First, we binarized the original 49 eSDMs maps for the analysis (1 current and 48 future climate). We transformed the probabilistic maps obtained with help of *Biomod2* binary suitable/nonsuitable maps using the threshold maximizing the TSS [55]. Afterwards, we evaluated the range shift under GCMs and climate change scenario for the selected time period by comparing the binary eSDM under current climate conditions (1970–2000) with the binary eSDMs obtained for a specific period (2021–2040, 2041–2060, 2061–2080, 2081–2100) using three metrics—gain, loss, change. The gain metric assesses the percentage of area (number of pixels-locations) acquisition that was not used in the current climate. The loss metric characterizes the percentage of area loss under the new climate. The change metric is equal in value to gain − loss, and characterizes the percentage of area change under the new climate. These estimates were made using the function of *Biomod2_RangeChange*. This function allows us to estimate the proportion of area (relative number of pixels) lost, gained, or stable for the time interval considered in the modelling of species range dynamics under conditions of global climate change. Comparative analysis of gain, loss, and change metrics for different GCM groups (Hsens, Msens, and Lsens), scenarios (SSP1-2.6, SSP2-4.5, SSP3-7.0, SSP5-8.5), and time periods (2021–2040, 2041–2060, 2061–2080, 2081–2100) was performed using a three-factor analysis of variance (ANOVA) with fixed effects.

We constructed SDMs using R [104] and R packages *Ape* [105], *Biomod2* [89], *Raster* [106], *Ecospat* [107], *ENMeval* [108], *ENMtools* [68], *SpThin* [64], and *Usdm* [69]. In addition, we applied R scripts presented in the literature [101] to assess the suitability of models using RSTUDIO v. 1.4.1106 software [109]. Visualization of the eSDMs was carried out in the ArcGis Desktop 10.8.1 environment [57].

## 3. Results

### 3.1. Characteristics of Records, Selected Predictor Variables, and Training Area

After applying the spThin subsampling procedure and the sequential removal of clustering records, 78 records remained, for which the ANNI = 1.09, i.e., there is a random distribution of records (z-value = 1.59; *p*-value = 0.11; observed average distance = 292 km, expected average distance = 267 km). The full clustered records (z-value = −27.96; *p* << 0.01), the reduced records (DSred), and the iSDMs training area are presented in Figure 1.

The distribution of *A. agrarius* records in the ecological space defined by the first two PCA axes is presented in Figure 2. This figure shows that the directions of the scattering ellipsoid axes for the full and reduced records are almost identical. In addition, the major axis of the ellipsoid (Figure 2B) is approximately parallel to Bio_19 (Figure 2C), and the minor axis to Bio_05 (Figure 2C). For this reason, the first two predictor variables for *A. agrarius* are Bio_19 (precipitation of coldest quarter, mm) and Bio_05 (maximum temperature of warmest month, °C). In Figure 2, these variables are highlighted in blue. Bioclimatic variables Bio_01 (annual mean temperature, °C), Bio_02 (mean diurnal range of temperature, °C), and Bio_12 (annual precipitation, mm) were also included in the set of predictor variables because all pairwise Spearman rank correlation coefficients were less than 0.72 in absolute value, and VIF coefficients for all variables were less than 5 (Appendix A). 

### 3.2. Predictor Variables for Creating SDMs

The analysis of CMIP6 GCMs allowed the selection of three groups of high-, medium-, and low-sensitivity (Hsens, Msens, Lsens) GCMs to study biological invasions in Northern Eurasia. Below is a brief analysis of this group of GCMs and scenarios of climate change in the 21st century. This is important, because we wanted to understand how much warming and what range shifts of the SFM are expected in the 21st century, compared to the 20th century. It is currently known that a warming of 0.7 °C in the 20th century has already caused the shift in ranges of many species by 6.1 km in a decade [15]. Our estimates show that the temporal patterns of temperature and precipitation changes averaged across groups of GCMs and scenarios have similar features, but they differ in magnitude (Appendix A). For example, if the average warming for GCMs and scenarios reaches 1.5 (±0.2) °C by 2021–2040 compared to the baseline of 1970–2000, then within the three groups of Hsens, Msens, Lsens GCMs, the warming by 2021–2040 is 1.66 (±0.15) °C, 1.53 (±0.17) °C, and 1.30 (±0.17) °C, respectively (Figure 3, Appendix A). The maximum level of warming is expected at the end of the 21st century (2081–2100). For all GCMs and scenarios, on average, the temperature will increase by 4.97 °C for the Hsens GCMs. For the Msens and Lsens GCMs, the warming level is lower, − 4.76 (±2.21) °C and 4.03 (±1.86) °C, respectively. The estimates show that the mean diurnal range of temperature (Bio_02) decreases against the background of warming, the value of which takes on the greatest value for the Hsens GCMs (Appendix A, Hsens). At the end of the 21st century (2081–2100), Bio_02 for the Hsens, Msens, and Lsens GCMs is expected to decrease by 9% (−1.26 °C), 4% (−0.35 °C), and 1% (−0.09 °C), respectively. These GCMs show increases by 37% (Hsens), 34% (Msens), and 5.9 °C (32% Lsens) in the maximum temperature of the warmest month of the year (Bio_05) at the end of the century (2081–2100). In contrast to temperature (Bio_01, Bio_02, Bio_05), the difference in precipitation (Bio_12, Bio_19) is less significant for different groups of GCMs. Estimates have shown that at the end of the 21st century (2081–2100), the increase in total annual precipitation (Bio_12) and total precipitation in the coldest quarter of the year (Bio_19) are characterized by the following values: for Hsens GCMs: Bio_12 – 14% (62 mm), Bio_19 – 20% (12.4 mm); Msens GCMs: Bio_12 – 14% (62 mm), Bio_19 – 14% (9.0 mm), and Lsens GCMs: Bio_12 – 11% (47 mm), Bio_19 – 12% (7.6 mm) (Appendix A).

The difference between the warming rates of the average annual temperature (Bio_01) and the total amount of annual precipitation (Bio_12) leads to a shift in the aridity index (I_DM_) of de Martonne [75], i.e., there is a decrease in the degree of moistening of the territories, and aridification of the entire Russian territory (Appendix A). For example, if under current climate conditions I_DM_ = 8.3, then the index value by 2021–2040, 2041–2060, 2061–2080, and 2081–2100 for GCMs and scenarios will be 6.86, 5.94, 5.27, and 4.8, respectively.

### 3.3. Optimal Parameters of iSDMs

The optimal iSDM parameters obtained from the *Biomod2_tuning* function are shown in Table 1. This table also lists the default model parameters. The dependence of the AUC value on the parameters of the GAM, GBM, FDA, RF, ANN, and MaxEnt models are shown in Appendix A. From these figures, we can see that for the GAMs with optimal parameters, predictive power improves by 0.13, i.e., the AUC metric with default parameters is 0.86 (AUC_def_ = 0.86), and with optimal parameters it is 0.99 (AUC_opt_ = 0.99) (Appendix A). The improvement of the AUC for the GBMs is significantly less (0.03) than GAMs, i.e., AUC_def_ = 0.94 and AUC_opt_ = 0.97 (Appendix A). For FDA models, the predictive power improvement in terms of AUC is 0.08, i.e., AUC_def_ = 0.9 and AUC_opt_ = 0.98 (Appendix A). For RF models, the improvement is the same as for FDA models, i.e., AUC_def_ = 0.915 and AUC_opt_ = 0.985 (Appendix A). For ANN, the quality improvement of the models is 0.09, i.e., AUC_def_ = 0.89 and AUC_opt_ = 0.98 (Appendix A). The improvement of the MaxEnt models with optimal parameters in terms of AUC is 0.07, i.e., AUC_def_ = 0.91 and AUC_opt_ = 0.98 (Appendix A).

### 3.4. Variable’s Importance in Created SDMs, Bioclimatic Niche Analysis, and Variable Response Curves

Violin plots of the variable’s importance are shown in Appendix A. This figure shows that for the two regression models (GLM, GAM), the most important variable is Bio_01 (VarI = 0.94 ± 0.006). For machine-learning models (ANN, GBM, RF), the importance of this variable is not significantly lower (VarI = 0.89 ± 0.04). For the FDA and MaxEnt models, the significance of this variable is VarI = 0.86 ± 0.01 and VarI = 0.87 ± 0.02, respectively. For the eSDM ensemble model, the importance of variable Bio_01 has an intermediate value (VarI = 0.90 ± 0.01). The second important variable for all models is Bio_05 (Appendix A). The second and third important variables in all models are Bio_05 and Bio_19, respectively (Appendix A). The next important variables differ little in the models, but in eSDM the importance of variables Bio_02 and Bio_12 differ significantly (VarI = 0.09 ± 0.01 and VarI = 0.18 ± 0.02) (Appendix A). Overall, a multiple analysis of the mean importance values of the variables in the eSDM using the Tukey HSD criterion showed (F = 4612, *p* << 0.01) that the importance of the variables differed significantly from one another. 

The bioclimatic niche of *A. agrarius* can be described by the tolerance of environmental factors at which the species can survive. Figure 4 shows the distribution of records along five of the environmental gradients making key contributions to the models. The tolerance of species for each predictor variable was estimated using the full set of *A. agrarius* occurrence records in the native and invasive parts of the range (Figure 4). The centroid of the niche of *A. agrarius*, in terms of average annual temperature, occupies a position in the region of positive temperatures (4.1 ± 3.83 °C), which differs significantly from the average annual temperature (−4.82 °C) in Russia (t = 93.3, *p* << 0.01). Niche centroids with regard to the other variables have a low coefficient of variation (CV = 17.8%) of average diurnal range of temperature (9.7 ± 1.83 °C), a high value of the maximum temperature of the warmest month (23.8 ± 1.83 °C) at a low value of CV (13.2), a relatively high value of CV (33%) of total annual precipitation (534 ± 178, mm), and rather high values of CV (=58%) of total annual precipitation in winter (85 ± 49, mm).

Although the response curve plots for the five variables in the GLM, GAM, GBM, FDA, RF, ANN, and MaxEnt models were constructed from a reduced set of records, they correctly identify the tolerance zones of the species to environmental factors (Appendix A). In summary, these response curves for variables Bio_05 (maximum temperature in summer) and Bio_19 (precipitation in winter) show that the species prefers habitats that are characterized by summer temperatures from 16 to 30 °C, and winter precipitation in the range from 10 to 200 mm. The response curves for Bio_12 show that the annual precipitation ranges from 250 to 750 mm. The response curves for Bio_01 and Bio_02 show that the ranges of variations are −1 to 10 °C and 8 to 15 °C, respectively.

### 3.5. Predictive Performance of SDMs

The predictive power assessments of iSDMs, as determined by the three metrics TSS, AUC, and KAPPA, are presented in Figure 5. The mean values of the TSS, KAPPA, and AUC metrics for all iSDMs obtained across all methods are 0.90 ± 0.05, 0.76 ± 0.08, and 0.97 ± 0.02, respectively. We will consider the assessments of the models’ accuracy using the TSS metric as an example, since it is recommended to use with a threshold value of 0.8 to create ensemble models. Using this threshold value to create ensemble models ensures that only “excellent” iSDMs will be used to create an ensemble model (see Section 2.5.2). Although the average value for TSS (Figure 5A) is quite high, 0.90, there is one implementation each from the FDA (TSS = 0.758), ANN (TSS = 0.763), and RF (TSS = 0.8) models, which have TSS metrics less than or equal to 0.8. It follows that, with a threshold value of TSS = 0.8, 60 iSDMs, i.e., 95% of all 63 models, will be taken into account when creating an ensemble model.

The predictive power of ensemble models created using two aggregation strategies (CA, WM) of iSDMs in term of TSS, KAPPA, AUC, and Boyce metrics is presented in Figure 5. There is an improvement in the estimates of all TSS, AUC, and KAPPA metrics obtained by both the CA and WM strategies. If the average of the TSS metric for iSDM was 0.90 ± 0.05 (Figure 5A), then for ensemble models, these metrics were higher: for the CA strategy–TSS-CA = 0.94 ± 0.0005, and for the WM strategy TSS-WM = 0.94 ± 0.004 (Figure 5D). At the same time, the dispersion of estimates decreased by more than 16 times. We can see from the figure that the quality scores of the eSDMs also improved on the KAPPA metrics (Figure 5B,D); for iSDMs it was 0.76 ± 0.08, for eSDMs on the CA strategy KAPPA-CA = 0.78 ± 0.002, and for the WM strategy KAPPA-WM = 0.77 ± 0.007. In addition to the increase in accuracy, there also are 40 and 11 times decreases in variance for the CA and WM strategies, respectively. The same significant improvements are observed for the AUC metric (Figure 5C,D): for iSDM-AUC = 0.97 ± 0.02, for eSDM by CA-strategy AUC-CA = 0.99 ± 0.0005, and by WM strategy AUC-WM = 0.99 ± 0.0005. 

We additionally assessed the quality of the ensemble models using the Boyce index (B_ind_) (Figure 5D), which showed rather high B_ind_ for both CA and WM strategies: Boyce-CA = 0.98 ± 0.0009, and Boyce-WM = 0.98 ± 0.001. Subsequently, to build ensemble models for various group of GCMs and scenarios of climate change in the 21st century, we chose the CA strategy to combine models, because the variance of predictions by this strategy is less than the WM strategy.

### 3.6. Potential Habitat Suitability of the SFM under Current Climatic Conditions

The eSDM created for current climate conditions shows that for the SFM, climatic suitability is higher in the central and south of European Russia, in the south of Western and Central Siberia, and in Russian Far East regions. Lower suitable ones are located in the gap zone in Transbaikalia, in the north of the European part of Russia, including the Republic of Karelia and the Arkhangelsk Region (Figure 6). The map (eSDM) also indicates suitable regions for the SFM in western Kamchatka, although this species has not yet been found in these regions.

### 3.7. Assessment of Species Range Shifts under Global Climate Change

Below are the results of a comparative analysis of the impact of different GCMs (Hsens, Msens, Lsens) and scenarios (SSP1-2.6, SSP2-4.5, SSP3-7.0, SSP5-8.5) on the SFM range shift in 2021–2100 with a step of 20 years. A comparative analysis of the estimates of gain metric of suitable areas shows that changes in terms of this index over time depend on the GCM groups (F = 151.8, *p* << 0.001, R^2^ = 99.6%, Figure 7A, Appendix A). The figure shows that while gain in the initial period in 2021–2040 does not differ for all groups of GCMs, for Lsens GCMs, this metric statistically significantly differs (*p* < 0.05) from Msens and Hsens GCMs for 2061–2080. A statistically significant difference in terms of gain for Msens and Hsens GCMs is observed only at end of the 21st century (2081–2100). The averaged values of the gain across the GCMs show that the difference in this metric across the four scenarios begins to differ particularly markedly (F = 384.5; *p* < 0.01) from the period 2061–2080 (Figure 7B). Estimates of the influence of different factors show that the gain of new areas is influenced by both the main factors (models, scenarios, years) and their interaction (*p* < 0.05) (Appendix A). In contrast to gain, loss of areas is weakly expressed (F = 2.43, *p* = 0.03, R^2^ = 79.6%), since a statistically significant difference is detected only for the main factors (models: F = 3.83, *p* = 0.04; scenarios: F = 7.75, *p* = 0.02; years: F = 6.14, *p* = 0.005) (Appendix A). The interaction of these factors is not statistically significant (*p* > 0.05, Appendix A). The average area loss for the Lsens, Msens, and Hsens GCMs is 7.2% (±0.12) (Figure 7C,D).

Overall assessments of the change index show that statically significant differences are revealed both by the main factors (models: F = 140.2, *p* << 0.01; scenarios: F = 403.2, *p* << 0.01; years: F = 888.3, *p* << 0.01), and interactions of factors (models × scenarios: F = 3.70, *p* = 0.01; models × years: F = 14.4, *p* << 0.01; scenarios × years: F = 78.0, *p* << 0.01) (Appendix A, Figure 7E,F). This is because these changes are more related to the gain of new areas, and a shift in range from south to north and from west to east (Figure 8, Figure 9 and Figure 10). The average values of changes in the areas according to all GCMs and scenarios of the species in Russia by 2081–2100 is 86.4% (±0.5). For the group of Lsens, Msens, and Hsens GCMs, the changes in the species’ range area by 2081–2100 are 75.2% (±0.8), 89.3% (±0.8), and 94.8% (±0.8), respectively. The ordering of changes in the areas according to the SSPx-y scenarios (SSP1-2.6, SSP2-4.5, SSP3-7.0, SSP5-8.5) shows that the largest value is achieved for SSP5-8.5 (107.8%), and the smallest for SSP1-2.6 (64.2%), i.e., the most aggressive climate change scenario (SSP5-8.5) results in 1.7 times more area change compared to the moderate scenario (SSP1-2.6). For other scenarios, SSP2-4.5 and SSP3-7.0, these metrics are intermediate, and amount to 80.5% and 93.4%, respectively. A temporal comparative analysis of the change in the area of the species range shows that by 2081–2100, the change (117%) will be 2.2 times higher than in 2021–2040 (53%). Changes in the range area by 2041–2060 (76%) and 2061–2080 (100%) will increase by 1.4 and 1.9 times compared to 2021–2040, respectively.

Changes in the potential distribution of *A. agrarius* from 2021 to 2100 under the conditions of three groups of CMIP6 GCMs and four SSPx-y scenarios of global climate change are shown in Figure 8, Figure 9 and Figure 10. These models predict that global warming will expand potentially suitable areas for *A. agrarius*. Moreover, these maps show that the centroid of species range will move from south to north and from west to east (Figure 11). Outcomes show that the centroid shift of range for Hsens is 753 (±9) km, while for other Msens and Lsens it is 670 (±9) km and 615.4 (±9) km, respectively. It is important to note that the shifts of the range centroids from west to east (Slong) and from south to north (SLat) differ significantly (Hsens: Slong = 720 km, Slat = 289 km; Msens: Slong = 621 km, Slat = 295 km; Lsens: Slong = 576 km, Slat = 258 km), i.e., the shift of the centroid by longitude is two times more than by latitude (Figure 11).

These maps show that area loss under various climate change models and scenarios happens mainly in the steppe (Volgograd, Astrakhan, Saratov, and Orenburg regions), as well as the mountainous areas of the Dagestan Republic (Figure 8, Figure 9 and Figure 10). Stable suitable areas are confined to the south of western and central European Russia, the southern regions of the Urals and Siberia, and the south of the Russian Far East.

## 4. Discussion

### 4.1. Model Considerations

A comprehensive consideration of the effects of various aspects (selection and/or definition of randomly distributed records, uncorrelated predictor variables, training area and optimal parameters of iSDMs, strategy and number of generation pseudoabsence points, SDM quality assessment metrics, method of combining iSDMs, thresholds for map binarization) of modelling is necessary for improving predictions of spatial patterns of biological invasions under conditions of global climate change. Currently, the above-mentioned aspects of modelling are among the important, but still insufficiently resolved issues of environmental modelling. Nevertheless, we followed best modelling practices [62,63,64,65,66,67,68,90,91,92,93,94,95,96,97,110,111,112,113,114,115,116,117] and created acceptable eSDMs with high values of Boyce indices (0.98 ± 0.0009). There are many different scoring measures to determine the accuracy and predictability of SDMs, such as the AUC, KAPPA, TSS, Boyce, or the Jaccard and Sorensen indices [55,66,67,102,112]. In our study, we selected AUC, KAPPA, and TSS to calibrate iSDMs. Although these methods are widely used in ecological modelling, some studies showed that AUC, KAPPA, and TSS may give misleading measures of model performance due to their dependence on prevalence [66,67,112]. For this reason, we additionally used the Boyce index to evaluate the accuracy of eSDMs. Final verification of the constructed eSDMs under current climate conditions using the Boyce index is important because it requires only records data, and guarantees the reliability of projecting the eSDMs in space and time under conditions of global climate change. Overall, SDMs are a useful tool in determining how the SFM will change its current range under future climate change. However, it is also clear that there are some limitations to SDMs in predicting the invasive spread. Geographic shifts in species range involve multiple ecological factors, such as dispersal, invasion pathways, physiology, between-species and interpopulation interactions, and evolution, operating at multiple scales [39,41,55,56]. Correlative SDMs cannot explicitly account for the effects of these factors, which interact with ecological processes, and may ultimately cascade to influence the invasive process [94]. Furthermore, there are other potential issues associated with the exclusion of topographic and landscape variables from models that can influence model outcomes [102,103]. In addition, future land use change scenarios (e.g., road building, deforestation for agriculture, expansion of agricultural fields) can also alter future species distributions. Improvements of SDMs based on long-term monitoring and ecological data, as well as increasing the dataset of occurrence records, are critical issues for enhancing the predictive accuracy of the models. However, we believe that the results we obtained are ecologically meaningful and mostly in accordance with our field survey data [36,38,56]; thus, additional consideration of other factors and alternative modelling options should not deviate significantly from the results presented in this study.

### 4.2. Why Are Selected Variables Important for the Creation of iSDMs and eSDMs?

Our analysis showed that among the variables that determine the iSDMs and eSDMs, four variables are the most important—Bio_01 (annual mean temperature), Bio_05 (max temperature of warmest month), Bio_12 (annual precipitation), and Bio_19 (precipitation of the coldest quarter). The importance of these variables for the species is confirmed by numerous studies conducted in the native and invasive parts of the range [46,118,119,120]. Rodent reproductive activity is regulated by temperature, precipitation, and the availability of sufficient food [119]. The temperature of the environment can have a strong influence on the reproductive rates of *A. agrarius*. For example, on agricultural land in China, the proportion of pregnant females increased with seasonal warming. At an average monthly temperature of 10 °C, it was 20%, and at 15–24 °C it was 60% [120]. However, hot summers (average monthly temperature ≥ 29 °C) reduced the rate of reproduction, which led to a bimodal pattern of population growth [120]. In the Primorsky Territory Nature Reserve in Russia, the litter size decreased in dry years. This confirms the hypothesis that acute thermal stress can have a negative impact on survival and reproduction [121,122]. 

Our estimates of average positions (centroids) of ecological niches along environmental gradients show a good agreement between the literature and our data. The evaluation of the centroid of the realized niches in terms of the Bio_05 variable showed that the maximum temperature of the warmest month (23.8 ± 1.83 °C) is significantly less than 30 °C (Figure 4C). The maximum temperature in summer is also consistent with the response curves (Appendix A). Although they differ for all models, nevertheless, most of the curves of the Bio_05 variable in the GLM, GAM, GBM, RF, and MaxEnt models show the optimal temperature range in summer—from 16 to 29 °C. Only for two FDA and ANN types of iSDMs, the optimal range is not much wider—from 15 to 30 °C.

Westward expansion of *A. agrarius* in Central Europe, which was observed at the turn of the 20th and 21st centuries, is associated with an increase in summer and winter temperatures [121]. Numerous studies in the western part of the range (Bulgaria, Russia) show that *A. agrarius* is more associated with higher moisture than other representatives of its genus [38,47,120]. In regions with precipitation less than 500 mm/year, the habitat preferences of the SFM become narrower, and it is found there mainly in humid habitats [23]. The assessment of the realized niche centroid in terms of variable Bio_12 showed that the sum of annual precipitation is not significantly greater than 500 mm/year (534 ± 178, mm). This estimate is also consistent with the response curves for the Bio_12 variable in the GLM (Appendix A), GAM (Appendix A), GBM (Appendix A), RF (Appendix A), ANN (Appendix A), and MaxEnt (Appendix A) iSDMs. From these curves, we can see that the optimal range of annual precipitation is from 250 to 750 mm. 

Overall, our study suggests that temperature (Bio_01) influences the survival and reproduction of *A. agrarius* [118,119,120,121]. Of course, other environmental variables, such as Bio_08 (mean temperature of wettest quarter, °C) and Bio_10 (mean temperature of warmest quarter, °C), can also effect *A. agrarius* reproduction, but these were not used because of the high Spearman correlation coefficient (Pspear) between Bio_01 and these variables (Pspear > 0.72, Appendix A). Frequent or severe changes in temperature can also have negative effects, so the variable of mean diurnal range of temperature was used in model construction (Bio_02) [42]. It follows from both our and the literature data that Bio_12 precipitation also influences species survival and reproduction [23], so this variable was also used to construct the SDMs. While the importance of variables Bio_01, Bio_02, Bio_05, and Bio_12 agrees with the published data [46,118,119,120,121], the importance of variable Bio_19 is due to the presence of the winter precipitation threshold. On the one hand, the small amount of snow in winter provides thermal conditions for the SFM to overwinter in their burrows. On the other hand, rapid snowmelt in spring prepares them for the breeding season to come. Based on our results, we can suggest a set of environmental drivers that control the distribution of *A. agrarius* in Northern Eurasia.

### 4.3. Specific Features of the Potential Range of the SFM at the End of the 20th Century

The SFM is a common species and its distribution is relatively well-studied [23,24,26,37,38]. However, the current distribution of the species needs to be constantly adjusted, as the mouse keeps invading different regions [36]. The potential range of the SFM (eSDM) in Russia, based on seven iSDMs (GLM, GAM, GBM, RF, FDA, ANN, MaxEnt) under the current climate conditions (Figure 6), is in good agreement with the known records, including confirming the presence of the European–East Asian range gap.

The European–Eastern Asian range gap can be traced in vertebrates of different classes (amphibians, reptiles, birds, mammals) [123,124]. This gap is revealed at different levels of taxonomic differentiation of western and eastern forms. For species that spread to the west, the gap often lies between 100° E and 110° E in Transbaikalia [125]. The range gap of the SFM fits into this range gap, and almost until the end of the 20th century it extended from Baikal to the upper reaches of the Amur River (Figure 6). At the same time, the differentiation of the SFM from the western and eastern parts of the range does not reach the subspecies level [39,41]. The range gaps began to be discussed as early as the 19th century, largely because paleontologists and zoogeographers discovered East Asian species in Europe [125,126,127]. In the last 50 years, the emergence of the European–East Asian range gap is more often associated with the events of the glacial period (more than 12 thousand years ago) [124]. The SFM ventured westward during one of the interglacial periods at the end of the Pleistocene, and the range gap may have occurred 38-9 thousand years ago [39,40]. It is important to point out that the pathogens of the SFM inhabiting the west (the European–Siberian–Kazakh part of the range) and the east of Eurasia (Far East-China) are different. This is observed, for example, for hantaviruses and leptospiruses (see Introduction). In other words, westward-dispersing SFM did not carry with them the causative agents of the main infectious diseases that formed in east of Eurasia.

### 4.4. Expected Changes in Global Climate in the 21st Century

CMIP6 GCMs are used by scientists and policy-makers to interpret past and future climate change, and to determine appropriate policies and optimal responses to the dangerous effects of climate change. However, these models are characterized by considerable uncertainty. As we noted previously, the ECS metric ranges from 1.83 °C to 5.67 °C, which makes their 21st century predicted warming levels very uncertain. In recent years (2020–2022), studies have performed uncertainty reduction by testing for consistency between model estimates of warming levels in 1980–2021, derived from CMIP6 GCMs and real ERA5-T2m surface air temperature records [128,129,130]. Thirty-eight CMIP6 GCMs, divided into three groups according to ECS (low-ECS, 1.80–3.00 °C; medium-ECS, 3.01–4.50 °C; high-ECS, 4.51–6.00 °C), were used for comparative analysis [130]. The results showed that the GCMs with high- and medium-ECS agree poorly with the observed field data. Several other studies have also shown that high ECS values (ECS > 3.0) are not supported by observations [131,132,133,134]. However, the GCM groups with low ESC have been found to be fully compatible, at least globally. Nijsse et al. [132] concluded that the most likely ECS interval should be 1.9–3.4 °C, whereas alternative studies based on empirical data suggest that the actual ECS may be even lower, probably between 1 and 2.5 °C [135,136,137]. Despite the importance of this retrospective comparative analysis, however, the main findings are tentative and require further verification. It is not obvious that the poor agreement between the high- and moderate-sensitivity GCM estimates and experimental data for the 1980–2020 time period can automatically be extended to other future time intervals for the period 2021–2100. Given the large range of ECS, and subsequent broad GCM predictions, having information on which GCM to have more confidence in would be of great value for biological invasions. However, in the absence of accurate estimates, there is a need to use Hsens, Msens, and Lsens GCMs to account for major trends in climate change. For this reason, we used all three groups of GCMs in this study. Among them is a group of Lsens GCMs under the SSP1-2.6 scenario (Lsens GCMs-SSP1-2.6), which is in perfect agreement with the empirical data [130]. For these Lsens GCMs-SSP1-2.6 models, moderate warming (ΔT) is expected over the next decades of the 21st century, i.e., ΔT for periods 2021–2040 and 2041–2060 will be less than 2.0 °C (ΔT = 1.13 °C, ΔT = 1.89 °C, respectively) (Appendix A). The warming for the period 2061–2080 could be 2.15 °C, but after CO_2_ emissions are reduced to zero by about 2075, the temperature for the period 2081–2100 would be 1.93 °C (Appendix A).

### 4.5. What Conclusions Can Be Drawn from the eSDMs under Climate Change?

Our predictive eSDMs, taking into account different groups of GCMs and scenarios of climate change, show that the two parts of the range of the SFM may merge in the future. However, the continuous range starts forming at different times in different GMCs and scenarios. For example, if for Hsens GCMs, the gap loss occurs by 2061–2080 under the SSP2-4.5 scenario, then for the SSP3-7.0 and SSP5-8.5 scenarios, it happens even earlier, by 2041–2060 (Figure 8). For moderate-sensitivity (Msens) GCMs, this change becomes noticeable under the SSP2-4.5 scenario by 2081–2100, and under the SSP3-7.0 and SSP5-8.5 scenarios, by 2061–2080 (Figure 9). For the Lsens GCMs, the loss of the range gap is noticeable by 2061–2080 under the SSP3-7.0 and SSP5-8.5 scenarios (Figure 10). 

Although the connection of the two parts of the range is a long process, the gap in the range has already begun to shrink: at the end of the 20th century, the SFM was found in Southeastern Transbaikalia within the territory that had been a range gap for this species for more than a millennium [138,139]. It is difficult now to predict the consequences of the merging of the western and eastern parts of the range, including the evolution of the SFM pathogens. However, it should be remembered that the dispersal of mice from the eastern part of the range, which is currently observed and predicted in the future, could trigger the spread of a severe form of hantavirus infection that is dangerous for humans. These results lead us to consider that the expected changes in the global climate, even under the most realistic scenario of moderate warming (GCM Lsens SSP1-2.6), can lead to the formation of a continuous range not only of the SFM, but also of other species with a European–East Asian range gap, such as muskrat (*Ondatra zibethicus*) and mink (*Neogale vison*) [140].

Range dynamics of the species also show that in some parts of the steppe zone, regardless of GCM sensitivity and climate change scenarios, the range of the SFM will decrease in the future. This is due to the difference in the rate of change in temperature and precipitation. Our climate change assessments (Appendix A) show that by 2021–2040, 2041–2060, 2061–2080, and 2081–2100, the temperature will increase by 30%, 56%, 81%, and 103%, but precipitation will increase by only 6%, 9%, 11%, and 13%, respectively, i.e., there is a decrease in the aridification index for the whole of Russia (Appendix A). This conclusion is consistent with the decrease in the de Martonne aridity index [141] (Appendix A). Although, for different GCMs and scenarios, these indices differ from the means, but the general patterns remain (Appendix A). For example, for the group of the most realistic Lsens warming GCMs, the de Martonne index decreases for the periods 2021–2040, 2041–2060, 2061–2080, and 2081–2100 are 7.02, 6.13, 5.55, and 5.18, respectively, i.e., the index decreases will be 15%, 26%, 33%, and 38%, respectively. For this reason, in the steppe zone, the aridity of the current habitats of the SFM will increase, natural habitats will become unsuitable for the species, and irrigation is unlikely to be economically profitable, i.e., such habitats will be lost [142]. Global climate change will affect the forest-steppe and forest zones of the habitat to a lesser degree. The eSDMs also show that in the north of the Asian part of Russia, the low-suitability habitat remains largely unchanged, while the medium (moderate)-suitability habitat in the south of the Asian part of Russia and in the northwest of the European part increases. Highly suitable habitats in central European Russia and the Far East remained unchanged. Overall, our results indicate that future habitats will be predominantly located in warm, humid, flat or low-mountain environments, with the northern regions, including the Kola Peninsula, and eastern Northern Eurasia, including Kamchatka, being at the highest risk of invasion.

### 4.6. Differences in the Impact of GCMs and Scenarios on the Dynamics of Range Change

The location and area of habitat suitable for the SFM vary across GCMs (Hsens, Msens, Lsens) and SSPx-y scenarios, indicating increasing uncertainty about the pattern and rate of its distribution under climate change. To understand the invasion processes of the species, and to identify the most sensitive invasion regions, one cannot be limited to a single GCM and scenario of global climate change. Differences between alternative SSPx-y scenarios are mainly related to changes in greenhouse gas concentrations, especially the effect of CO_2_ concentration on temperature [71,72]. Climate change under the SSP1-2.6 scenario or, in the worst case, SSP2-4.5, could slow down the expansion of the SFM as much as possible. Additionally, an increase in the high consumption of fossil fuels (SSP3.70, SSP5-8.5) will lead to a clear expansion of the species to the northwest. Thus, sustainable development will not only protect our environment, but also limit the spread of invasive organisms and their subsequent naturalization.

## 5. Conclusions

In this study, we used an ensemble approach, with the application of seven different algorithms, bioclimatic variables, and species occurrence records in the native and invasive parts of the range, to modelling the spatial distribution of *A. agrarius* under current and future climate changes in Russia. Mean annual temperature, mean monthly temperature variation, maximum summer temperature, total annual precipitation, and total precipitation in the coldest period of the year play key roles in its reproduction and overwintering. Our analysis highlights that global climate change may further extend the invasive range of *A. agrarius* to the northeast, and transform the range gap of the SFM (and possibly other species with similar distribution), which has existed for the past 12 thousand years, into a continuous range from Central Europe to East Asia, including various countries of northern Eurasia. Our results provide an important scientific basis for organizing SFM population limitation measures, and for predicting the distribution of this species in the context of global climate change.

## Figures and Tables

**Figure 1 biology-12-01034-f001:**
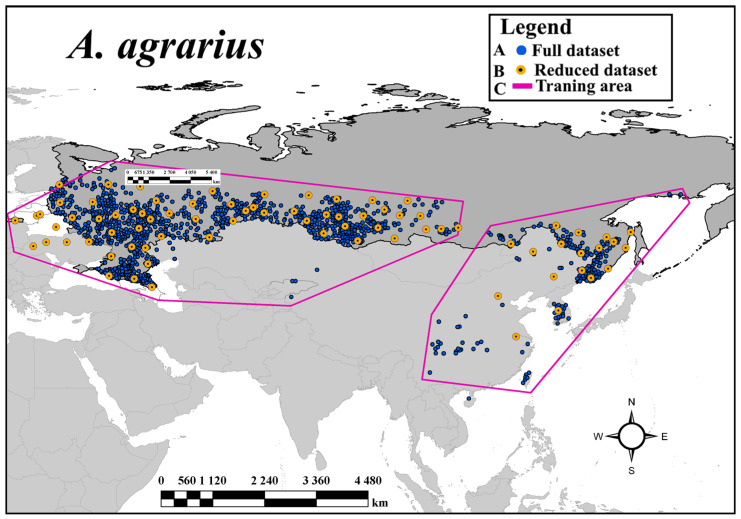
Locations of the records and SDM training area based on the available data sets in Eurasia, where A is the initial full clustered dataset, B is the reduced random distributed dataset, and C represents masks (training area) used to calibrate iSDMs.

**Figure 2 biology-12-01034-f002:**
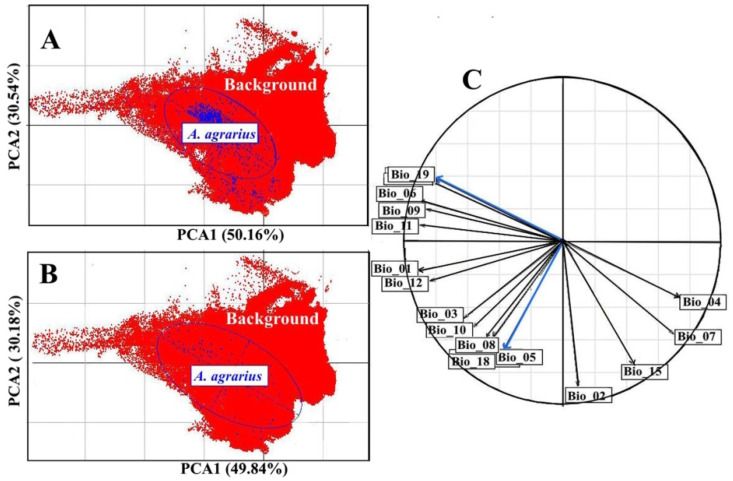
Distribution of *A. agrarius* records (**A**,**B**) in the ecological space defined by the first two PCA axes, and (**C**) correlation circle showing the correlations between bioclimatic variables (see Appendix A). In the panels (**A**,**B**), the red color represents the background points, the blue dots represent the full (**A**) and the reduced records (**B**), respectively, and the blue lines represent the scatter ellipsoids of the full and reduced records. The blue arrows in the panel (**C**) are selected variables using PCA.

**Figure 3 biology-12-01034-f003:**
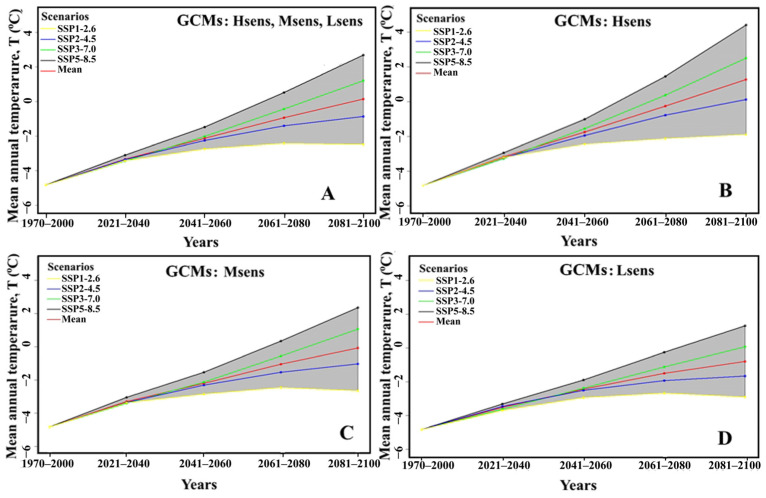
Assessments of average annual air temperature in the 21st century based on all (**A**) and different groups of GCMs, Hsens (**B**), Msens (**C**), and Lsens (**D**), under different SSPx-y climate change scenarios in Northern Eurasia.

**Figure 4 biology-12-01034-f004:**
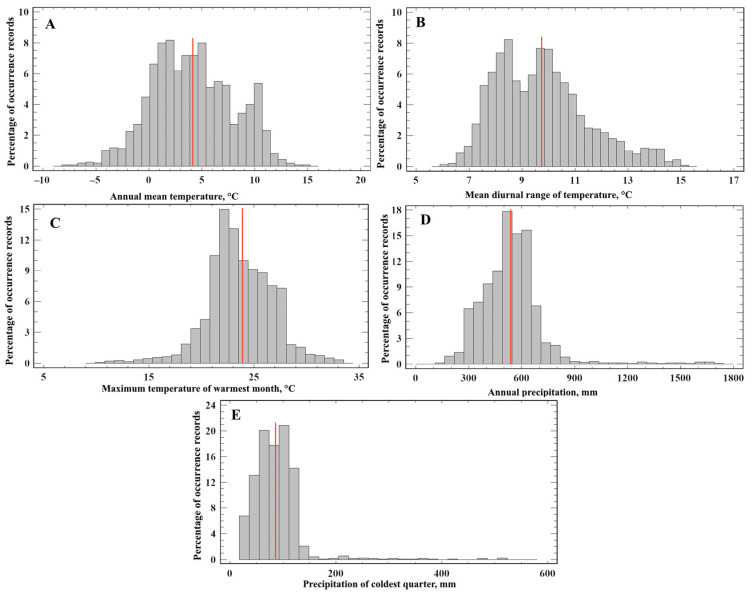
Histogram of *A. agrarius* occurrence along a gradient of (**A**) annual mean temperature, (**B**) mean diurnal range of temperature, (**C**) maximum temperature of the warmest month, (**D**) annual precipitation, and (**E**) precipitation of coldest quarter. Red lines indicate average positions (centroids) of ecological niches along environmental gradients.

**Figure 5 biology-12-01034-f005:**
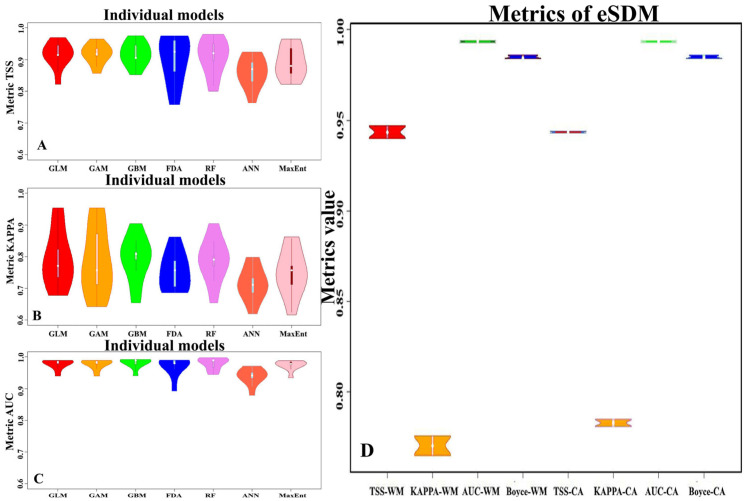
Violin plots of predictive power of iSDMs and eSDM constructed by two strategies of iSDM aggregation WM (TSS-WM, KAPPA-WM, AUC-WM, Boyce-WM) and CA (TSS-CA, KAPPA-CA, AUC-CA, Boyce-CA), in panels (**A**–**C**) presented TSS, KAPPA, AUC metrics values for iSDMs respectively, and in the panel (**D**) presented TSS, KAPPA AUC, Boyce metrics values of eSDMs for both strategies (WM, CA) of aggregation.

**Figure 6 biology-12-01034-f006:**
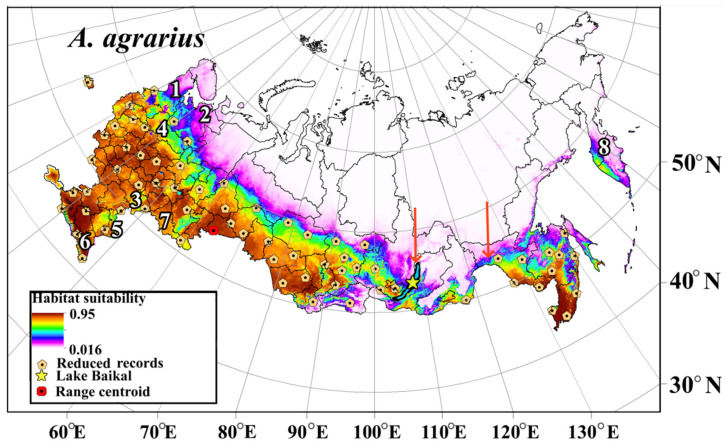
Potential range of the SFM built using eSDM adapted under current climate conditions. The red arrows indicate the longitudinal boundaries of the disjunction of the range, for the period before the 1990s. The numbers indicate the subjects of Russia mentioned in the text: 1—Republic of Karelia, 2—Arkhangelsk Region, 3—Saratov Region, 4—Volgograd Region, 5—Astrakhan Region, 6—Dagestan Republic, 7—Orenburg Region, 8—Kamchatka Territory.

**Figure 7 biology-12-01034-f007:**
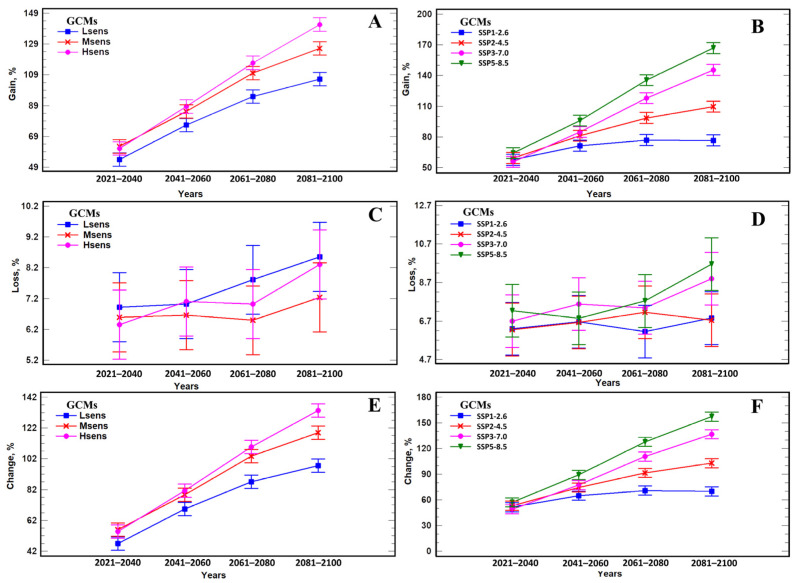
Comparative analysis of gain, loss, and change metrics of *A. agrarius* areas under three groups of GCNs and four scenarios of global climate change using three-way analysis of variance (ANOVA) with fixed effects. Means are presented with Tukey’s 95% confidence intervals. In the panels (**A**,**C**,**E**) presented Gain, Loss and Change metrics values respectively, depending on GCM sensitivity (Hsens, Msens, Lsens), and in the panels (**B**,**D**,**F**) presented Gain, Loss and Change metrics values respectively, depending on climate change scenarios (SSPx-y).

**Figure 8 biology-12-01034-f008:**
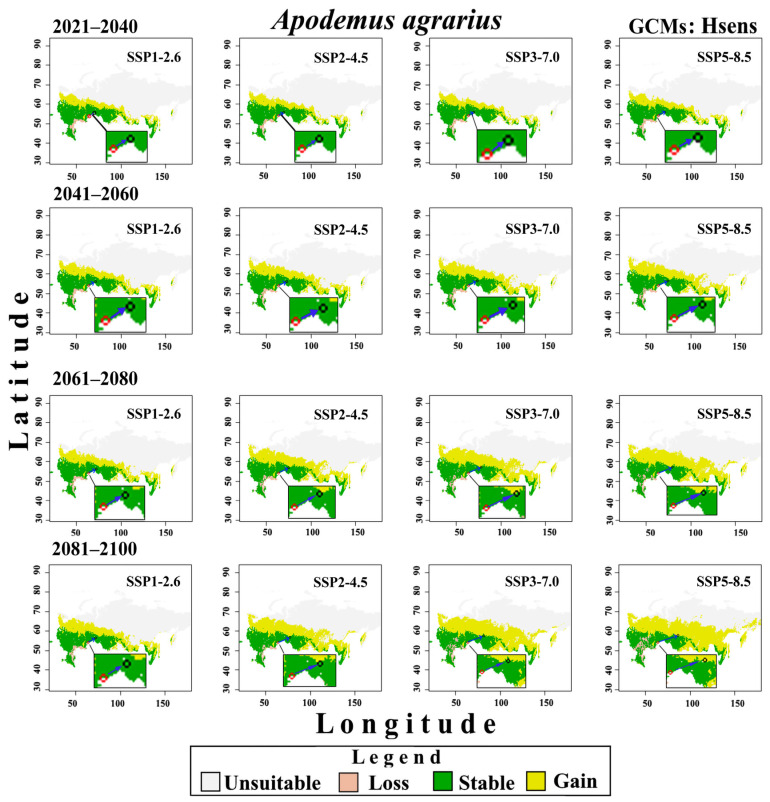
Changes in the potential distribution of *A. agrarius* in Russia from 2021 to 2100 under Hsens GCMs and four scenarios SSPx-y of global climate change. The red and black circles indicate the centroids in the current (1970–2000) and under four climate change scenarios in the future, the blue arrow indicates the direction of the centroid’s shift.

**Figure 9 biology-12-01034-f009:**
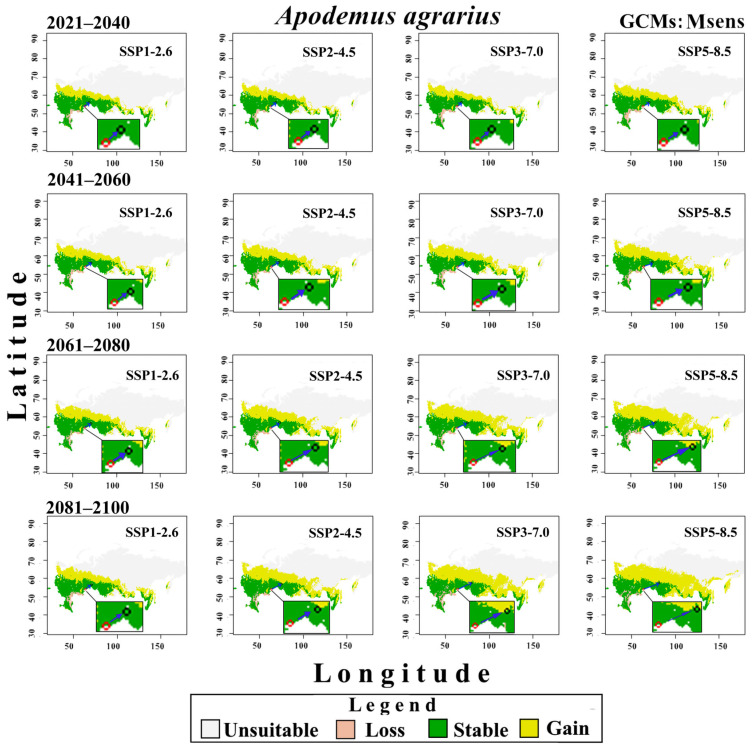
Changes in the potential distribution of *A. agrarius* in Russia from 2021 to 2100 under Msens GCMs and four scenarios SSPx-y of global climate change. The red and black circles indicate the centroids in the current (1970–2000) and under four climate change scenarios in the future, the blue arrow indicates the direction of the centroid’s shift.

**Figure 10 biology-12-01034-f010:**
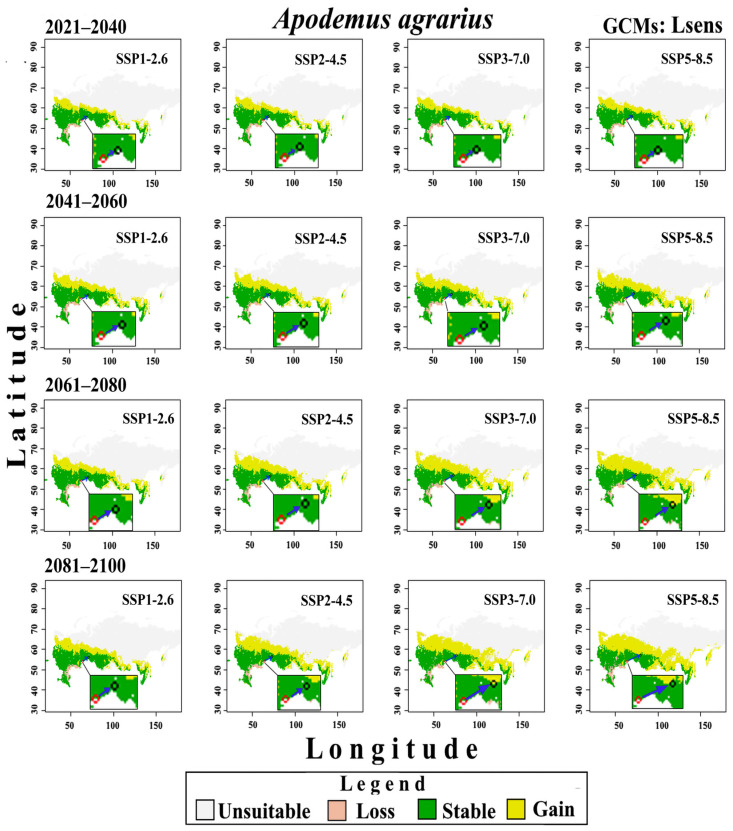
Changes in the potential distribution of *A. agrarius* in Russia from 2021 to 2100 under Lsens GCMs and four scenarios SSPx-y of global climate change. The red and black circles indicate the centroids in the current (1970–2000) and under four climate change scenarios in the future, the blue arrow indicates the direction of the centroid’s shift.

**Figure 11 biology-12-01034-f011:**
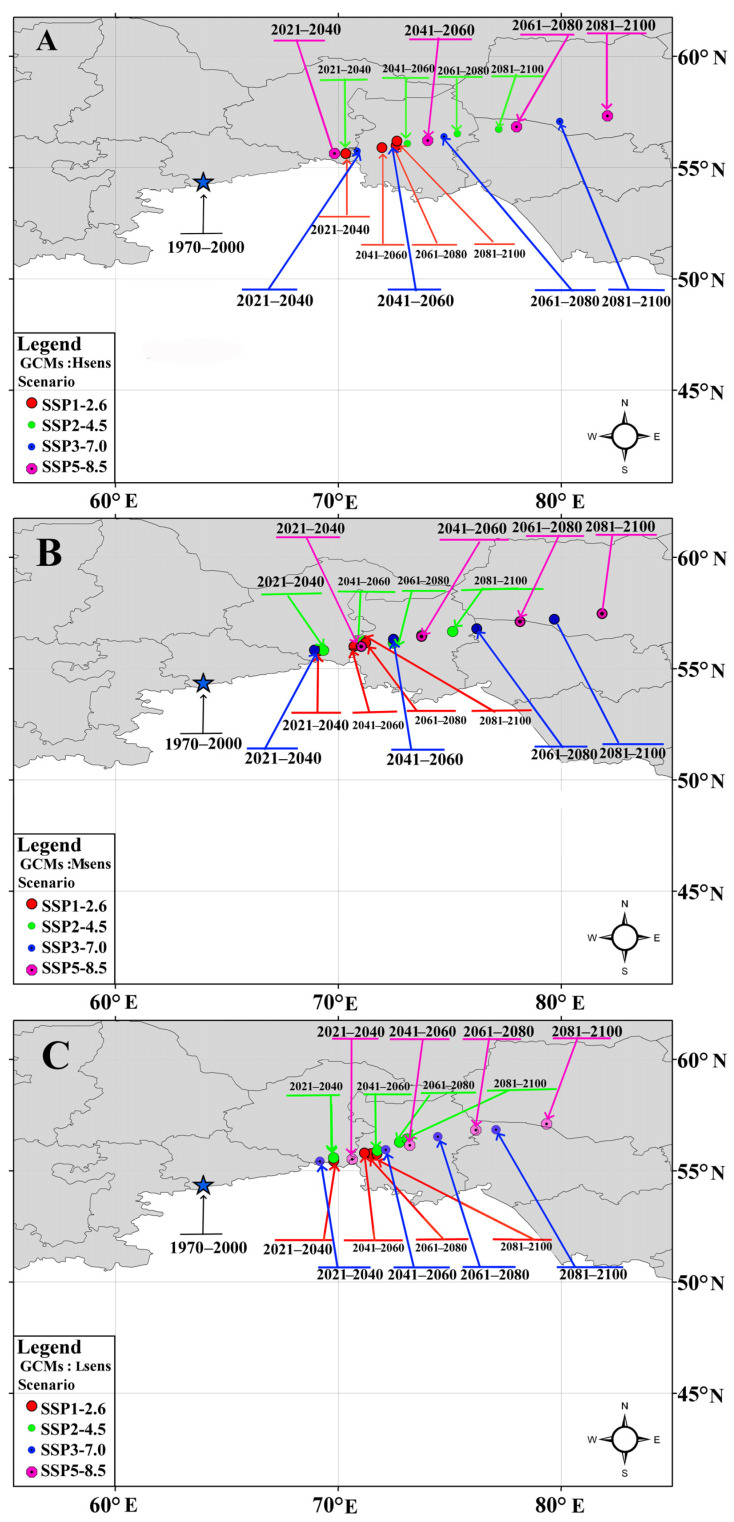
Geographical distribution of *A. agrarius* potential range centroids from the late 20th century to 2100s under groups of (**A**) Hsens, (**B**) Msens, and (**C**) Lsens GCMs and four scenarios SSPx-y of global climate change. The geographic position of the centroid of the potential range under current climate is indicated by blue star.

**Table 1 biology-12-01034-t001:** Optimal parameters of iSDMs (GLM, GAM, GBM, FDA, RF, ANN, MaxEnt).

iSDM Methods	Default Parameters	Optimal Parameters
GLM	type = ‘quadratic’; interaction.level = 0; myFormula = NULL	type = ‘quadratic’; interaction.level = 0; myFormula = *A. agrarius*~Bio_01 + I(Bio_01^2^) + Bio_02 + I(Bio_02^2^) + Bio_05 + I(Bio_05^2^) + Bio_12 + I(Bio_12^2^) + Bio_19 + I(Bio_19^2^)
GAM	k = −1, interaction.level = 0; select = FALSE	k = 2, interaction.level = 1; select = FALSE
GBM	n.trees = 2500; interaction.depth = 7; shrinkage = 0.001	n.trees = 10,000, interaction.depth = 9, shrinkage = 0.0005
FDA	add_args = NULL (degree = 1; nprune = NULL)	degree = 2; nprune = 16
RF	ntree = 500; mtry = 4, nodesize = 5, maxnodes = NULL	ntree = 500; mtry = 2, nodesize = 5, maxnodes = NULL
ANN	size = NULL (=5); decay = NULL	size = 6; weight decay = 0.001
MaxEnt	Linear = TRUE; Quadratic = TRUE; Product = TRUE; Threshold = TRUE; Hinge = TRUE, RM = 1	Linear = TRUE; Quadratic = TRUE; Product = FALSE; Threshold = FALSE; Hinge = TRUE, RM = 4

## Data Availability

Data are contained within the article or Appendix A.

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
