# Peer review of "Range Dynamics of Striped Field Mouse (Apodemus agrarius) in Northern Eurasia under Global Climate Change Based on Ensemble Species Distribution Models"

_biology, 2023, doi:10.3390/biology12071034_

Round 1
Reviewer 1 Report (Previous Reviewer 2)

Author Response
Dear reviewer!
We are very grateful for your valuable suggestions and comments, which have enabled us to further improve the quality of our MS. We hope that we have answered all the questions and comments and clarified all dubious places.
General Comments
1.1. Introduction: I appreciate the authors’ efforts in framing their study into a broader context of biological invasions but have some more suggestions for improvement.
RESPONSE 1.1. Thank you for your great understanding and positive assessment of our study
1.2. Some of the content in the first two paragraphs could be streamlined to be more specific to the impacts of climate change on disease-spreading invasive species, since that is what the present study focuses on.
RESPONSE 1.2. Done (Lines 50-57)
1.3. For example, in the first paragraph, can you give more specific examples that are relevant to disease-spreading species? -
RESPONSE 1.3. Done (for example, brown (Rattus norvegicus) and black (R. rattus) rats (Lines 53-57).
1.4. In the second paragraph, the content could focus more on disease-spreading invasive species in northern Eurasia, with specific examples of how climate suitability models (risk maps) can help with their surveillance and management (with citations).
RESPONSE 1.4. Done. We believe that this is enough (Lines 50-57)
1.5.The third paragraph could then introduce SFM. However, I recommend switching the 3rd and 4th paragraphs, so that the threat of SFM is introduced before details are given about the history of its invasion.
RESPONSE 1.5. Done (Lines 70 -87).
1.6. More clarification is needed as to why climate is expected to strongly shape the distribution of SFM. As written, it seems like human land-use change is the major range-limiting factor. Some info from the Discussion on this topic could be summarized (lines 744-752).
RESPONSE 1.6. Corrected. (Lines 88-111)
2. I think there is a misunderstanding about my previous comment related to model extrapolation. The reason that I recommended conducting a MESS or MOP analysis didn’t have anything to do with niche conservatism. Projecting a model to new conditions (e.g., a future climate change scenario) may lead to unreliable predictions because climatic variable values for future conditions may fall outside of the range values found in values used for model fitting, and variables may exhibit different correlation structures. You want to avoid interpreting model predictions in areas where strict extrapolations occur. I realize the paper already has many tables, figures, and analyses in general; however, I do this this is important. Below are some references to review regarding this issue.
Elith, J.; Kearney, M.R.; Phillips, S.J. The art of modelling range-shifting species. Methods Ecol. Evol. 2010, 1, 330−342. https://doi.org/10.1111/j.2041-210X.2010.00036.x.
- Charney, N.D.; Record, S.; Gerstner, B.E.; Merow, C.; Zarnetske, P.L.; Enquist, B.J. A test of species distribution model transferability across environmental and geographic space for 108 western North American tree species. Front. Ecol. Evol. 2021, 9, 1–16. https://doi.org/10.3389/fevo.2021.689295.
- Owens, H.L.; Campbell, L.P.; Dornak, L.L.; Saupe, E.E.; Barve, N.; Soberón, J.; Ingenloff, K.; Lira-Noriega, A.; Hensz, C.M.; Myers, C.E.; et al. Constraints on interpretation of ecological niche models by limited environmental ranges on calibration areas. Ecol. Modell. 2013, 263, 10–18. https://doi.org/10.1016/j.ecolmodel.2013.04.011.
RESPONSE 2. We agree that this issue is important. As you previously mentioned, we have a large number of tables and figures, and additional analysis would increase the material, which will make this article difficult for readers. Other reviews suggested to reduce the number of figures to make the essence of the study more clear. For this reason we have moved one figure from the main text to the supplementary material. Of course, we will use the suggested methods in our work in the future.
3. Discussion/Conclusions:
- Some parts of the Discussion could be streamlined and re-organized.
3.1.The Conclusions section reads more like an abstract but should instead summarize the most important points of the study.
RESPONSE 3.2. We have shortened and rewritten the conclusion as indicated in your suggestion (Lines 1038-1051)
- 3.3. Very little discussion is devoted to implications of the study, such as the spread of diseases carried by SFM.
RESPONSE 3.3. We have added to existing sections (Lines 902-907, 957-962).
- 3.4. The previous version of this manuscript highlighted the role of land cover in shaping SFM’s distribution. In the Discussion, can you speculate on how future land use changes may interact with climate change to increase risk of invasion?
RESPONSE 3.4. Unfortunately, according the literature and our own data, there areno specific results on land use, and adding such a section would be speculative.
4. A lot of the details regarding selection of optimal parameter values in “Section 2.5.1” could be included as Supplementary Material.
RESPONSE 4. Done (Lines 308-387).
5. Please reduce the number of abbreviations for words – it makes the paper harder to read. For example, I don’t recommend abbreviating “bioclimatic variables” to “BVs” (you could just shorten it to “variables”), “background points” to “BPs,” or “pseudo-absence points” to “PA.” The word “records” could be used instead of “SORs.”
RESPONSE 5. Done.
6. Grammar: check the manuscript carefully for sentence run-ons, comma splices, and other incorrect absences of punctuation. I notice that there is some improper use of the word “the” throughout the paper. For example, a “the” should go before “potential distribution” throughout the paper (e.g., “We studied the potential distribution”), whereas “the climate change” should just be “climate change.” Define all abbreviations in captions throughout the paper and SI.
RESPONSE 6. Done.
7. I think all country boundaries should be shown in the relevant figures (e.g., Figs. 7, 12). They could be shaded a different color (e.g., gray) if they’re not relevant to the study.
RESPONSE 7. Done (Figures 1, 11).
Specific Comments
8. L76: The reader needs more context for the relevance of this paragraph, so I’d start with a sentence that gives an overview of its invasion history (e.g., “SFM has undergone a significant range expansion over the past XXX years, with the most recent invasions occurring in XXX”). Then go into more details.
RESPONSE 8. Done (Lines 92-93)
9.L105: “Northern” should be lowercase (“northern Eurasia”). The study is not really “predicting invasions”, but just predicting which areas may be suitable for establishment under current and future climates.
RESPONSE 9. Done (Line 139)
10. L119: Cite the date of download of the GBIF data (e.g., “accessed 1 Jun 2023”). It looks like you only used one published study, so “literature sources” should just be “literature source.”
RESPONSE 10. Done (Lines 154-155)
11. L128: I don’t understand the difference between SOR dataset #1 vs. #3 – i.e., what is meant by “we used only SORs” for #3?
RESPONSE 11. For the first type of records we know the exact geographic coordinates, and for the third type of data we know only the names of the locality, which we used to determine the geographic coordinates using GoogleEarth (https://earth.google.com).
12. L135: There appears to be a typo in the spatial resolution (it should be ~5km2).
RESPONSE 12. Agree. Spatial resolution is ~ 5 km x 5 km (Line 172)
13. L164: This last sentence could be removed (it’s unclear what “thin out BVs” even means).
RESPONSE 13. Done (Line 215)
14.L166: “select variables from the full WordClim 2.1 dataset” reads better.
RESPONSE 14. Done(Line 203)
15.L170: “published recommendations” is too vague – can you provide a couple of specific examples, briefly?
RESPONSE 15: Done (Lines 207- 215)
16.L176-179: I’m not completely understanding this process. PCA is usually conducted to reduce the dimensionality of correlated variables, and the SDM is then estimated using PC1, PC2, etc. Here it appears that two bioclimatic variables were selected based on how close they were to the PC1 and PC2 axes. From a biological standpoint, this doesn’t make a lot of sense. Were those two particular variables more influential on SFM’s survival or development than other variables that were in a similar position in the multidimensional space (maybe explained in the Discussion, L773-790)?
RESPONSE 17. PCA approach to selecting variables to construct SDMs has been proposed Guisan, A.; Thuiller, W.; Zimmermann, N.E. “Habitat suitability and distribution models. Cambridge University Press: Cambridge, United Kingdom, 2017”. They suggest selecting the first two variables that are not too collinear (the two variables pointing in orthogonal directions are independent) and contribute significantly to the overall environmental variation. In selecting the remaining variables, we relied on expert knowledge of the ecology of the species and the study region. It is important to note that the proposed formal way of selecting the first two variables is universal in that it can be used for any species occurrence records and a full set of bioclimatic variables. Of course, we then tested against the literature why these variables are important for the creation of iSDM and eSDM (see Discussion 4.2). Of course, we then checked with the literature why these variables are important for the construction of iSDM and eSDV (see Discussion 4.2).
18. L189: replace “impact” with “potential impact”
RESPONSE 18. Done (Line 233)
19.L221-225: This info doesn’t really belong in the Methods. Also, species distribution modeling doesn’t always use correlations (e.g., there are mechanistic species distribution models).
RESPONSE 19. This paragraph has been removed(Lines 264-268). But with some changes, it is included in the introduction section at the suggestion of the second reviewer (Lines 122-136).
20. L232: Ensemble models do not always exhibit higher performance. See Hao et al. (2020) https://doi.org/10.1111/ecog.04890 for a review on this topic (I also recommend citing this study). At a minimum, “demonstrate” should be replaced with “may demonstrate.”
RESPONSE 20. Done (Lines 1333, ref 91).
In addition, there are special studies that showed that “The top method was an ensemble of tuned individual models. In contrast, ensembles built using the biomod framework with default parameters performed no better than single moderate performing models.”
https://esajournals.onlinelibrary.wiley.com/doi/pdf/10.1002/ecm.1486
21. L232: What is meant by “a wide range of approaches for model validation?” Combining multiple models into an ensemble isn’t validation. Model validation would involve collecting an independent record dataset to confirm that the individual models or ensemble model is accurately predicting the distribution, for example.
RESPONSE 21. Of course we agree with you. This was a mistake. We corrected (Lines 275-276).
22. L244: This sentence should be reworded. The default parameter values aren’t always appropriate, not the parameters themselves.
RESPONSE 22. We meant “tuning parameters”. Corrected (Line 288)
23.L251: “doParallel” (typo). Provide citations for R packages. I recommend using italics or quotation marks for names of R packages and functions.
RESPONSE 23. Done (Line 295). The revised MS version uses italics for the names of R-packages and functions.
24.L268: I think “liner” should be “linear.”
RESPONSE 24. Done. This section included in the Supplementary materials.
25. L276: Sentence run-on – need a punctuation mark here.
RESPONSE 25. Done. This section included in the Supplementary materials.
26.L283: Unclear what result is considered “satisfactory.” Can you be more specific?
RESPONSE 26. This means that the mgcv R-package cannot determine the optimal values for the two parameters by default. Corrected. This section included in the Supplementary materials.
27. L346: This should be stated earlier.
RESPONSE 27. Yes, it's stated earlier (Lines 280-285) .
28. L348: Species in the training area? This doesn’t make sense.
RESPONSE 28. Corrected (Line 392).
29. L350: Repeated sentence. Cite the references for the “Sre” generation strategy.
RESPONSE 29. Done (Line 396).
30. L354: Optimal parameter values identified in the previous step. The parameters weren’t defined, just the values.
RESPONSE 30. Corrected (Line 399).
31. L357: Should “predictive efficiency” be “predictive performance?” This is a cross-validation method, right?
RESPONSE 31. Yes. Corrected (Line 402)
32. L361: Replace “productive” with “predictive” (this typo also occurs elsewhere).
RESPONSE 32. Corrected(Line 407).
33. L367: Methods described in this paragraph should be cited. L380 should be re-worded. A low correlation indicates that the variable is important to the model.
RESPONSE 33. Everything is correct here. The text explains how the importance of a variable is evaluated in Biomod2 by using the formula VarI = 1 - Pcor. If Pcor takes high values, it means that randomized variable has little effect on the prediction (1 - Pcor) and is considered not important for the model. Here, two predictions are compared, the reference and the new prediction that is built after the randomization of one variable. We have added a reference (Line 415).
34. L389: Unclear what is meant by “created in parallel.” I’d delete this.
RESPONSE 34. Deleted (Line 437-438).
35. L398: How exactly was the best model selected? Best score across all three metrics?
RESPONSE 35. We agree. Not very specific. The best model is selected by the maximum value of the TSS metric. Corrected (Lines 446-447).
36. L409: This sentence could be deleted. It’s not clear if you’re talking just about the Boyce index or all metrics together.
RESPONSE 36. Deleted (Line 458-459)
37. L420: A sentence is needed here to describe why the results are binarized (to estimate the potential distribution or range).
RESPONSE 37. Since Biomod2 evaluates the Gain and Loss metrics by comparing pixel values under current and new climate conditions, it is therefore necessary to determine pixel characteristics as suitable/unsuitable for raster maps under current and new climate conditions. Without binarization, the Biomod2 package cannot determine the values of the above metrics. We think that part of the text regarding the definition of Gain and Loss metrics is presented in detail in MS (Lines 469-487).
38. L427: I don’t know that “habitat” is an appropriate term here. You’re estimating changes in the potential distribution based on climate. So, the Gain metric would assess increases in climatically suitable areas under future conditions. If the word “habitat” is used, it should be clarified that you’re defining this as climatically suitable areas.
RESPONSE 38. Agree. Gain is formally defined as - the number of suitable pixels (locations) predicted to be gained, i.e. this area. We replaced "habitat" with "area" (Lines 489-494)
39. L445: The first few sentences seem to be methods – please reword or remove this information so that only results are presented.
RESPONSE 39. Corrected (Lines 503-507).
40. L449: Why are there two “<” symbols?
RESPONSE 40. In this case Pvalue=0.00000001, to note that a Pvalue of much less than 0.01 is usually used double symbol “<”, i.e. P<<0.01(Line 507).
41. L452: You reduced bias, but the distribution of points is not random.
RESPONSE 41. We think this is correct since we determined it using the ArcGIS Desktop Pro 10.6.1 special function “Mean Nearest Neighbor” (see section 2.3). The selected records are randomly distributed within the minimum bounding rectangle that includes the points.
42. L477: The importance of all the information in this paragraph with respect of modeling SFM in needs to be summarized.
RESPONSE 42. Done. We hope that we summarized as you suggested (Lines 539-542).
43. L557: It’s unclear why there is a focus on temperatures in Russia. Can the nice centroid be pointed out in Figure 5?
RESPONSE 43. Done. Unfortunately, in this work we study range expansion only in Russia (now is Figure 4, Line 635).
44. L565: What is meant by the “correctly identify the tolerance zones?” How do you know what is correct?
RESPONSE 44. We found that the response curves plotted in the iSDM models (S14-S20) do not differ significantly overall from the diagrams (Figure 4) that were obtained using all records (raw data). It can be seen that the response curves correctly identify the tolerance zones of the species to environmental factors.
45. L618: “Suitable” is relative – it should be described as lower vs. higher (e.g., suitability is lower in the gap area, but it’s not zero in most areas) in this paragraph. Content in the Discussion (around L885) is relevant here.
RESPONSE 45. Corrected (Lines 681-684).
46. L634: A reminder is needed either here or somewhere in this paragraph why the results are important, or how they relate to the study overall.
RESPONSE 46. Corrected (Lines 699-701).
47. L733: No, the Boyce index doesn’t guarantee reliability of SDM projections. This is something that a MOP or MESS analysis could provide insight into though.
RESPONSE 47. We assume that since the Boyce index values (0.98 ± 0.0009) are high enough and the constructed response curves (Figures S14-S20) in the individual models (model zone of species tolerance to environmental parameters) agree relatively well with the full realized niche of the species derived from all occurrence records (Figure 4), a reliable SDM projection is therefore guaranteed. This matching is important because models based on data covering only a limited part of a species geographical range may lead in truncated or biased response curves if geographical truncation also results in ecological truncation, which will introduce errors when projected the models. Obviously, there are some limitations, which have been further considered as suggested by the third reviewer (Lines 806-822).
48. L751: I think a different word than “opinion” is required here. The first sentence of the next paragraph also should be reworded (estimates of what?).
RESPONSE 48. Corrected (Line 838).
49. L755: Clarify that 30C is when reproduction drops off, and what is meant by “These data..”
RESPONSE 49. Agree. Here we mean “The maximum temperature in summer”. Corrected (Line 844).
50. L764: Maybe replace “moisture-loving” with “associated with higher moisture” or similar.
RESPONSE 50. We agree. Corrected (Line 853).
51. L774: The second sentence doesn’t really make sense.
RESPONSE 51. We agree. The second sentence has been removed (Line 863-865).
52. L781: Negative effects on SFM? Do you have a citation?
RESPONSE 52. We agree. Reference added (Line 871).
53. L792: This section has some interesting material, but I think more clarification is needed regarding its relevance to this study. Combining this section with material in Section 4.5 (L858) here would help. Could hybridization of the two isolated SFM subgroups potentially alter genetics of the viruses/bacteria that it carries?
RESPONSE 53. We agree. We have added a few sentences on the medical significance of the expansion of field mice in the east of its range (Lines 902-907, Lines 957-962). The issue of the gap for vertebrates in Transbaikalia has been discussed for more than 100 years. However, no hypotheses regarding the causes have been previously proposed. We have now discovered for the first time that this gap is of climatic origin. In another in our publication on the spatial distribution of two species muskrat (Ondatra zibethicus) and mink (Neogale vison) in the current climate, we found that there is such a gap.
54. L814: This paragraph could be shortened.
RESPONSE 54. We agree. A rather detailed discussion may be given here, but we think it is important for understanding what global warming can actually be expected in the 21st century. We ask that you leave this paragraph unshortened.
55. L880-884: Clarify what is meant by irrigation not being profitable and give citations. The models didn’t include forage resources so I’m not sure what the last sentence means.
RESPONSE 55. The last sentence is deleted (Lines 982-985).
Regarding the unprofitability of irrigation, we can say that such a problem exists in Russia. Research shows that from 1990 to 2018, the area of irrigated land in the steppe regions has significantly decreased, their ameliorative condition has worsened. The area where capital works are required to improve the technical level of irrigation systems has sharply increased. The area of actually irrigated lands is progressively decreasing. We have added a relevant reference (Line 982).
56. L903: The climate scenarios is not “implemented.” Maybe state “climate change under the SSP1-2.6 scenario could slow down…” (remove the word “will” – we don’t know for sure what will happen).
RESPONSE 56. We agree. Corrected (Line 1003-1005).
57. L909: The Conclusions should be much shorter and summarize the major points and implications of the study
RESPONSE 57. We agree. Done (Line 1038-1051).
58. Figure 1: Green and red is hard for color-blind folks, so I’d use a different color for the reduced record dataset. You may also consider shading land areas to distinguish them from ocean. “Calibration” should be “calibrate.”
RESPONSE 58. Done (Figure 1, Line 520).
59. Figure 2: The blue arrows in “C” need to be defined in the caption.
RESPONSE 59. We agree. Done (Figure 2, Line 527).
60. Figure 4: Briefly summarize how to interpret a violin plot and define all abbreviations in plots.
RESPONSE 60. We have changed the figure captions in more detail. Currently, violin plots are widely used in the literature and are similar to box-plots, except that they show the probability density of the data at different values. Typically, violin plots include all of the data that are in the boxplot. However, box-plots have limitations in displaying data because their visual simplicity obscures a number of important details about how the data values are distributed. For example, with box-plot you will not see whether the distribution is bimodal or multimodal. We think the interpretation of violin charts is not correct to include in the MS, since these plots are widely used in the literature (Figure S13, Line 202).
61.Table S2: Specify that the units are degrees Celsius.
RESPONSE 61. Done (Line 38)
62. Table S3: Define the “De Martonne Index” in the caption.
RESPONSE 62. Done (Line 59).
63. Figure S3: Try to fix the year ranges on the x-axis. Maybe make them diagonal so you can type out the entire year ranges.
RESPONSE 63. Done (Lines 65, 71, 76, 82).

Reviewer 2 Report (New Reviewer)
This article presents predictions of current and future climate suitability for an invasive rodent aiming to improve the control the spread of its populations. The authors performed extensive analyses. The study is well designed, and well described. I particularly appreciated that the authors were careful on the interpretation of the effect of their predictors, with a very good dedicated paragraph in that regard.
I fully agree with the conclusions of the article: “Our results provide an important scientific basis for organizing measures to limit the population of SFM and for predicting the distribution of this species in the context of global climate change “
but you need to adjust the terminology in many places. Although, this will not alter the conclusions, it is best to be accurate on what is done here so that the SDM user community, especially the invasion risk modellers, are well informed on the content of this study. I strongly recommend to be more accurate on formulations in general (I explain them after). SDM can do a lot of different things which are not interchangeable (e.g., ecological hypothesis testing, quantifying ecological niche, modelling distribution, climate suitability, habitat suitability, environmental suitability, invasion risk, projection to future conditions or not...).
General comments
About the term “distribution”: You are modelling the climate niche of the species. Modelling its “distribution” implies that you take into account habitat, dispersal ability and anthropogenic factors of spread (eg roads, ports). To better reflect what you are modelling, I suggest you replace “distribution” by climate suitability when referring to your aim and results.
The title does not reflect well the study: in my opinion this is an invasion risk assessment. I suggest you include the term “invasion” (or invasive species), “future” (or risk) and “projection” (or prediction). That is because “Range dynamics” seems to refer to observed variation in site occupancy, since no element refers to ‘future’ projections. Invasion risk assessments are highly valued, and I fear a lot of readers will miss this article with the present title.
You present a lot of figures, including some that are not very interesting compared to the rest of the results (e.g. fig 3, fig 6) which could be in supporting information. Also you show all individual projections for each GCM. You can present a single figure projection of all GCM combined (average between all), since the article emphasizes the use of “ensemble modelling”. We do not see much differences between GCMs, so I really think it is not worth showing them all individually. This is particularly important with regard to operators, who will use your study for intervention, but do not know which GCM is the best to take into account. For a better clarity and guidance, also better robustness, I suggest you average your projections between GCMs.
Introduction
The aim of this study introduced here ‘to assess the impact of climate change’ does not reflect what is really done. With that formulation, we expect a study of the observed effect of PAST climate change (i.e. climate change effects) on current distribution. What you are doing here is ‘predicting and projecting the effect of climate change’ on the species potential distribution (more accurately, on climate suitability). In other terms, you want to project invasion risks (based on climate only, because you do not take into account habitat and factors of spread).
I see you made extensive projections (4 scenarios, 4 periods etc…). To reach your aim, this may not be needed (one period and 2 scenarios would have been sufficient). Here, it seems that you wish to predict the spread of the species, considering this time series. The title of the article reflects more this aspect (range dynamics) except that you need to specify that this is for the future, and they are predictions and not observations.
Methods
For your information, there is a study that assessed the performance of several of the algorithms you chose:
https://esajournals.onlinelibrary.wiley.com/doi/pdf/10.1002/ecm.1486
As you can see, biomod ensemble has not the best performance. Also, this is just a slight refinement of the methods, because a result is more driven by the data (which is very good in your case) than by the choice of algorithm (which we can observed also in the aforementioned article). So I am not asking to reperform the analyses. But in the future, you can simplify your analyses and make them more robust by only keeping GAM, GBM and Maxent.
2.5.1 Determination of the iSDMs optimal parameters
I think you are here referring to predictor selection. Optimal parameters can refer to algorithm settings, and not the predictor composition of the model. I suggest you rename this subsection for more clarity and standard fitting.
2.6
Althout we keep seeing binarization of distribution projection, it is never recommended, even when we want to quantify a surface, or losses or gains in suitablity. It is an extra effort for a loss of information. That is most of the time because we do not know the threshold of suitability for a species. Although you used a recommended methods (maxTSS), you here treat regions with very high invasion risk equal to regions with moderate invasion risk. This is particularly misleading in the case of invasive species that are still spreading (and did not fill the potential environment available). I suggest to justify why doing a binarisation, or present either differences in suitability scores (quantitative) or you make categories (high / medium / low invasion risk, or explicitely show the distributions of invasion risks between regions; see for instance Lanner et al. 2022).
https://www.sciencedirect.com/science/article/pii/S0048969722013389
I think your best justification for this transformation is the assessment of shifts of the centroids. In that case, this is totally fine, but you need to inform the reader of what do represent the binary-transformed maps: is it low to high risk of invasion? Or is it only high risk of invasion? What does your maxTSS threshold represent in this gradient? But the binary maps are less informative in that case.
For your information, we always can represent these with quantitative scores, by showing the differences in suitability between present and future (future – present, and we still can see where the species may spread when the change is positive). Calculations of surfaces are also not better after binarization, even for computing surface critaria of the IUCN (see for instance Guillera-Arroita et al 2015).
https://onlinelibrary.wiley.com/doi/pdf/10.1111/geb.12268
Note that your map (fig 7) is excellent.
I hope it helps!
Good luck.
Author Response
Dear reviewer!
The authors are grateful for valuable comments which helped us to improve MS. All changes are summarized below. We hope that we have answered all the questions and comments and clarified all dubious places.
Point-by-point response
General comments
1.About the term “distribution”: You are modelling the climate niche of the species. Modelling its “distribution” implies that you take into account habitat, dispersal ability and anthropogenic factors of spread (eg roads, ports). To better reflect what you are modelling, I suggest you replace “distribution” by climate suitability when referring to your aim and results.
RESPONSE 1. Done. We agree that more correct terminology is needed. In our work, species distribution models (SDMs) are built using climate variables (Bio_01, Bio_02, Bio_05, Bio_12, Bio_19), so we have adjusted the study objectives in terms of "climate suitability" (Lines 138-148).
2. The title does not reflect well the study: in my opinion this is an invasion risk assessment. I suggest you include the term “invasion” (or invasive species), “future” (or risk) and “projection” (or prediction). That is because “Range dynamics” seems to refer to observed variation in site occupancy, since no element refers to ‘future’ projections. Invasion risk assessments are highly valued, and I fear a lot of readers will miss this article with the present title.
RESPONSE 2. We agree to change the title if the other two reviewers do not disagree. Invasion risk assessment of striped field mouse (Apodemus agrarius) in Northern Eurasia under global climate change based on ensemble species distribution models.
3.You present a lot of figures, including some that are not very interesting compared to the rest of the results (e.g. fig 3, fig 6) which could be in supporting information. Also you show all individual projections for each GCM. You can present a single figure projection of all GCM combined (average between all), since the article emphasizes the use of “ensemble modelling”. We do not see much differences between GCMs, so I really think it is not worth showing them all individually. This is particularly important with regard to operators, who will use your study for intervention, but do not know which GCM is the best to take into account. For a better clarity and guidance, also better robustness, I suggest you average your projections between GCMs.
RESPONSE 3.We agree that the main part of the article contains a lot of figures, so we moved one figure 4 to the supplementary material (Figure S13, Line 202). The same suggestion was made by second reviewer. Regarding the average values across all GCMs and SSPx-y scenarios, such a figure is available for the variable Bio_01 (average annual temperature) (Figure 3A, Line 567). For the high, moderate to low sensitivity models, a graph of the change in all variables is also presented in the supplementary material (Figure S3-S6, Line Table S3, Lines 65, 71, 76, 82). Recent studies show that the expected global climate warming is better described by the CMIP6 GCM Lsens models (see Section 4.4, Lines 909-942), therefore only the models within each group (Lsens, Msens, Hsens) are averaged. Averaging all models without consideration of model sensitivity is not recommended. A discussion of this issue is presented in some detail in Section 4.4, indicating which group of models (Lsens) is recommended for future use.
Introduction
4.The aim of this study introduced here ‘to assess the impact of climate change’ does not reflect what is really done. With that formulation, we expect a study of the observed effect of PAST climate change (i.e. climate change effects) on current distribution. What you are doing here is ‘predicting and projecting the effect of climate change’ on the species potential distribution (more accurately, on climate suitability). In other terms, you want to project invasion risks (based on climate only, because you do not take into account habitat and factors of spread).
RESPONSE 4. We agree with the comment. We have corrected the purpose of the work. Regarding PAST, we can say that there is an incorrect use of the term "historical". A third reviewer suggested that averaged data for the period 1970-2000 would be better called "current".
5.I see you made extensive projections (4 scenarios, 4 periods etc…). To reach your aim, this may not be needed (one period and 2 scenarios would have been sufficient). Here, it seems that you wish to predict the spread of the species, considering this time series. The title of the article reflects more this aspect (range dynamics) except that you need to specify that this is for the future, and they are predictions and not observations.
RESPONSE 5. We agree. This detailed analysis provided because we wanted to analyze the impact of different models and climate change scenarios on the centroids shift. This is a very important issue that is widely discussed in the literature.
Methods
6.For your information, there is a study that assessed the performance of several of the algorithms you chose:
https://esajournals.onlinelibrary.wiley.com/doi/pdf/10.1002/ecm.1486
RESPONSE 6. Thank you. We will use these results in our work in the future.
7. As you can see, biomod ensemble has not the best performance. Also, this is just a slight refinement of the methods, because a result is more driven by the data (which is very good in your case) than by the choice of algorithm (which we can observed also in the aforementioned article). So I am not asking to reperform the analyses. But in the future, you can simplify your analyses and make them more robust by only keeping GAM, GBM and Maxent.
RESPONSE 7. We agree that the results are very dependent on the quality of the data. In addition, there are special studies that showed that “The top method was an ensemble of tuned individual models. In contrast, ensembles built using the biomod framework with default parameters performed no better than single moderate performing models.”
https://esajournals.onlinelibrary.wiley.com/doi/pdf/10.1002/ecm.1486
8. 2.5.1 Determination of the iSDMs optimal parameters
I think you are here referring to predictor selection. Optimal parameters can refer to algorithm settings, and not the predictor composition of the model. I suggest you rename this subsection for more clarity and standard fitting.
RESPONSE 8. This means the setting of model parameters (2.5.1). The selection of variables is presented in another section (2.3, Lines 203 - 229). In this section, the description starts with the statement that the optimization is concerned with «tuning parameters of seven iSDMs». The distinction between «model parameters» and «environment variables» is widely presented in the literature.
9. 2.6. Although we keep seeing binarization of distribution projection, it is never recommended, even when we want to quantify a surface, or losses or gains in suitability. It is an extra effort for a loss of information. That is most of the time because we do not know the threshold of suitability for a species. Although you used recommended methods (maxTSS), you here treat regions with very high invasion risk equal to regions with moderate invasion risk. This is particularly misleading in the case of invasive species that are still spreading (and did not fill the potential environment available). I suggest to justify why doing a binarization, or present either differences in suitability scores (quantitative) or you make categories (high / medium / low invasion risk, or explicitly show the distributions of invasion risks between regions; see for instance Lanner et al. 2022).
https://www.sciencedirect.com/science/article/pii/S0048969722013389
RESPONSE 9. Thank you for the article. Very interesting research and suggestions. We are informed that in the literature there are different ways of presenting modelling results. However, the use of three category classes (high / medium / low) to create the final predictive maps is not well justified, as the selection of three thresholds to divide into three groups is not a simple challenge. Currently, a single maxTSS threshold is used mostly in the literature to construct the eSDM maps. In summary, our study uses two ways of presenting the modelling results: (1) under current climate conditions, climate suitable regions are presented in a continuous scale including (high / medium / low) categories (in the revised version - Figure 6), (2) predictive maps of climate change conditions are only presented using maxTSS, which depict regions at very high risk of SFM invasion. The use of maxTSS was suggested by two other reviewers.
10. I think your best justification for this transformation is the assessment of shifts of the centroids. In that case, this is totally fine, but you need to inform the reader of what do represent the binary-transformed maps: is it low to high risk of invasion? Or is it only high risk of invasion? What does your maxTSS threshold represent in this gradient? But the binary maps are less informative in that case.
RESPONSE 10. Yes, we agree with you that these maps reflect regions at high risk of species invasion, as we used maxTSS for binarization and TSS>0.8 metric values when combining individual models. Moreover, it is important that we additionally tested the predictive performance of ensemble models using Boyce index, the effectiveness of which is shown in the scientific literature. In the methodological part of the paper (Section 2.6), we specifically state that the centroid dynamics analysis is considered for binary maps. We believe that our approach more accurately reflects the main trends in centroid shifts over time.
11. For your information, we always can represent these with quantitative scores, by showing the differences in suitability between present and future (future – present, and we still can see where the species may spread when the change is positive). Calculations of surfaces are also not better after binarization, even for computing surface critaria of the IUCN (see for instance Guillera-Arroita et al 2015).
https://onlinelibrary.wiley.com/doi/pdf/10.1111/geb.12268
RESPONSE 11. Of course, we deeply understand the complexity of this issue. Yes, surely, this issue can be solved by various methods, but we decided to create climate suitability maps for the SFM using well-known methods of ensemble modeling (Biomod2) under global climate change (taking into account various models and scenarios) of the sixth generation of CMIP6. We choose «biomod2» because it is the most popular among ensemble SDM users and it is therefore representative of the typical ensemble SDM workflow (Hao et al. 2019, ref. 91). We carried out this detailed analysis because it is the approach that enables us to obtain niche shifts (centroids) of the SFM over time under climate change.
12. Note that your map (fig 7) is excellent.
RESPONSE 12. Thanks.
I hope it helps!

Reviewer 3 Report (New Reviewer)
Please see the attachment.

Author Response
Dear reviewer!
We are very grateful for your valuable suggestions and comments, which have enabled us to further improve the quality of our MS. We hope that we have answered all the questions and comments and clarified all dubious places.
The manuscript entitled " Range dynamics of striped field mouse (Apodemus agrarius) in Northern Eurasia under global climate change based on ensemble species distribution models” used distribution modeling approach (i.e., an ensemble modeling) together with relevant climate variables to predict near-current and future distribution pattern of an invasive species (striped field mouse). The manuscript is written well and the drawn conclusions are coherent with the obtained results. Although similar methodologies are common, results of the study could have useful implications for management actions. The manuscript requires some changes before its ready for publication:
Comments and suggestions:
Abstract
1. Line 28: “under historical (1970-2000)” in the context of SDMs this time period is often expressed as ‘current’ and sometimes as ‘ near-current’; please revisit this expression.
RESPONSE 1. We agree. We replaced "Historical" with "Current" everywhere in the revised MS.
Introduction
- 2.A new paragraph reviewing available modeling techniques is recommended.
RESPONSE 2. Done (Lines 122-138)
-3. Introduction could benefit from extra review of the target species in the context of this study, at the regional or global scale, if available.
RESPONSE 3. We corrected the introduction and added text in the context of your suggestion (Lines 47 - 109)
- 4.The justification text for the study is overstated. Could be shortened a little bit.
RESPONSE 4. We agree. However, in the previous version of the MS, another reviewer asked to give a detailed justification for our study.
Materials and Methods
-5.Line 116: “ArcGis” please revisit this, it is written as “ ArcGIS’
RESPONSE 5. Done (Line 492).
-6.Line 129: “an accuracy of at 129 least 5 km” please clarify what does that mean ?
RESPONSE 6. Since we used bioclimatic layers with a spatial resolution of 2.5 arc minutes (~5 km x 5 km), we would therefore like the occurrence records to be referenced with the same accuracy.
-7.Line 155-164: I suggest adding a reference to this technique.
RESPONSE 7. Done (Lines 192-202).
-8. Line 226-227: “In recent years, different methods, including regression, machine learning, classification, and maximum entropy, have been used to create SDMs [41,45,78,79].” Please refine this statement. Maximum entropy is a machine learning approach, isn’t it?
RESPONSE 8. Done (Line 270-284).
- 9.Line 236-240: Please see the above comment.
RESPONSE 9. Done (Lines 280-285).
-10.Line 350: “This procedure was repeated three times. This 350 procedure was repeated three times.” Please refine this expression.
RESPONSE 10. Deleted (Line 395).
-11. “2.5. Species distribution modelling” this section is quite wordy; I suggest shortening it. This is also true for the other sections following “2.5.”
RESPONSE 10. We agree. We partially moved section 2.5.1 to supplemental materials. However, for the previous version of MS, another reviewer made comments that the methodology is written without specificity. For this reason, please, leave these sections unshortaned.
Results
12. “Figure 1.” Please add a scale bar to this map.
RESPONSE 12. Done (Line 520)
13.“Figure 6, D” this should be presented with better resolution please.
RESPONSE 13. Done (now it is Figure 5, Line 669)
- 14. I suggest moving some of the figures to the supplementary document.
RESPONSE 14. Done. Violin plots of the variables importance moved to supplementary materials (Figure S13, Line 202).
Discussion
15. Line 724-725: “Nevertheless, we followed best modelling practices [41–724 47,72,75,76,79,81,85,86,89,91,98–100]” please revisit this expression and only use the most relevant references; too many reference for such an expression?
RESPONSE 15. Corrected (Line 794).
-16. Line 737: “4.2. Why selected variables are important for the creation of iSDMs and eSDMs?” This section requires more in-depth explanation on how climate variables used in model building contribute to the distribution of the target species.
RESPONSE 16. We have involved all the literature available to us to explain: how the climatic variables used in the model construction contribute to the distribution of A. agrarius. We hope that future studies on the ecology of this species will allow us to do this in more detail.
17. A small section highlighting the benefits of the applied modeling techniques in establishing priority zones for management actions is necessary.
RESPONSE 17. We did not include a new section, but the most important area (Transbaikalia) for management action we highlighted and discussed in detail in sections 4.3 and 4.5.We have added a few sentences on the medical significance of the expansion of striped field mice in the east of its range (Lines 902-907, Lines 957-962). The issue of the gap for vertebrates in Transbaikalia has been discussed for more than 100 years. However, no hypotheses regarding the causes have been previously proposed. We have now discovered for the first time that this gap is of climatic origin. In another our publication on the spatial distribution of two species muskrat (Ondatra zibethicus) and mink (Neogale vison) in the current climate condition, we found that there is such a gap.
- 18.Modeling limitations should be discussed within a section or subsection.
RESPONSE 18. Done (Lines 806-822).

Round 2
Reviewer 3 Report (New Reviewer)
The manuscript is sufficiently improved.
Best wishes,
This manuscript is a resubmission of an earlier submission. The following is a list of the peer review reports and author responses from that submission.
Round 1
Reviewer 1 Report
This manuscript focuses on an interesting topic; analyses are clear, and, overall, the interpretation of the results is sounds. I do see a few points, which I recommend to address:
1) Please provide a definition for considering SFM as an 'invasive species of North Eurasia'. Your data have evidently demonstrated expansion of are of distribution of this rodent species. But this does not characterize this species as invasive for Northern Eurasia. It was not introduced to this continent from another one. You can justify the definition in the text, but I recommend deleting the word 'invasive … in Northern Eurasia' from the title. Agree that the SFM is a native species for North Eurasia.
2) The second comment relates to the first one. The range of distribution of SFM was continuous before the European-East Asian gap was formed. In this case, the observed process of expansion of the distribution is more a restoration of the original range rather than 'invasion' to new territory. If paleontological paleo-climatic data suggest that the range gap has happened as the result of glacial events, it means that range expansion and shrinking processes are natural, not a result of the invasion.
3) On lines 79-80, you define the goal of this study to assess the impact of climate and land-use changes on distribution of SFM. Most land -use changes are caused by human activities (deforestation, agriculture, roads, etc.). However, I cannot see an analysis of the impact of these factors on the distribution of SFM.
4)I understand that all data on SFM occurrences are taken from GBIF. Can you discuss potential data collection biases in this regard. Specifically, my concern is about territories reoccupied by SFM recently. Can those data relate to a more extensive collection of rodents within these territories?
5) I recommend providing the full names of the most important variables in the discussion (line 509) and the table 1. For example, to use 'average annual temperature' instead of Bio01, etc. It would be much more interesting for readers to see the names of the variables, not the letters.
Reviewer 2 Report
Please see the attached file.
